# Role of serine/threonine protein phosphatase PrpN in the life cycle of *Bacillus anthracis*

**Aakriti Gangwal**[1], **Nitika Sangwan**[1ʘ], **Neha Dhasmana**[1ʘ], **Nishant Kumar**[1,2], **Chetkar Chandra Keshavam**[1], **Lalit K. Singh**[2¤], **Ankur Bothra**[3], **Ajay K. Goel**[4], **Andrei P. Pomerantsev**[3], **Stephen H. Leppla**[3]*, **Yogendra Singh**[1]*

**1** Department of Zoology, University of Delhi, Delhi, India, **2** CSIR-Institute of Genomics and Integrative Biology, Mall Road, Delhi, India, **3** Microbial Pathogenesis Section, Laboratory of Parasitic Diseases, National Institute of Allergy and Infectious Diseases, National Institutes of Health, Bethesda, Maryland, United States of America, **4** Division of Biotechnology, Defence Research and Development Establishment, Gwalior, Madhya Pradesh, India

ʘ These authors contributed equally to this work.
¤ Current address: InVitaGO Diagnostic GmbH, Frankfurt am Main, Germany.
\* sleppla@niaid.nih.gov (SHL); ysinghdu@gmail.com (YS)

**Data Availability Statement:** All relevant data are within the manuscript and its supporting information files.

## Abstract

Reversible protein phosphorylation at serine/threonine residues is one of the most common protein modifications, widely observed in all kingdoms of life. The catalysts controlling this modification are specific serine/threonine kinases and phosphatases that modulate various cellular pathways ranging from growth to cellular death. Genome sequencing and various omics studies have led to the identification of numerous serine/threonine kinases and cognate phosphatases, yet the physiological relevance of many of these proteins remain enigmatic. In *Bacillus anthracis*, only one ser/thr phosphatase, PrpC, has been functionally characterized; it was reported to be non-essential for bacterial growth and survival. In the present study, we characterized another ser/thr phosphatase (PrpN) of *B. anthracis* by various structural and functional approaches. To examine its physiological relevance in *B. anthracis*, a null mutant strain of *prpN* was generated and shown to have defects in sporulation and reduced synthesis of toxins (PA and LF) and the toxin activator protein AtxA. We also identified CodY, a global transcriptional regulator, as a target of PrpN and ser/thr kinase PrkC. CodY phosphorylation strongly controlled its binding to the promoter region of *atxA*, as shown using phosphomimetic and phosphoablative mutants. In nutshell, the present study reports phosphorylation-mediated regulation of CodY activity in the context of anthrax toxin synthesis in *B. anthracis* by a previously uncharacterized ser/thr protein phosphatase—PrpN.

## Author summary

Reversible protein phosphorylation at specific ser/thr residues causes conformational changes in the protein structure, thereby modulating its cellular activity. In *B. anthracis*, though the role of ser/thr phosphorylation is implicated in various cellular pathways including pathogenesis, till date only one STP (PrpC) has been functionally characterized.

**Funding:** Research reported in this manuscript was supported by SERB CRG grant (No. CRG/2018/000847/HS) and SERB JC Bose fellowship (No. SB/S2/JCB-012/2015) to YS, and in part by intramural funding from the National Institute of Allergy and Infectious Diseases, NIH to SHL. The funders had no role in study design, data collection and analysis, decision to publish, or preparation of the manuscript.

**Competing interests:** The authors have declared that no competing interests exist.

This manuscript reports functional characterization of another STP (PrpN) in *B. anthracis* and with the aid of a null mutant strain (BAS Δ*prpN*) we provide important insight regarding the role of PrpN in the life cycle of *B. anthracis*. We have also identified the global transcriptional regulator, CodY as a target of PrpN and PrkC, and for the first time showed the physiological relevance of CodY phosphorylation status in the regulation of anthrax toxin synthesis.

## Introduction

*B. anthracis*, a spore-forming bacterial pathogen is the causative agent of anthrax that majorly affects livestock, farm animals and sometimes humans [1–3]. It is a Gram-positive, aerobic, rod-shaped, chain-forming bacterium within the genus Bacillus with a mystifying life cycle determined by various environmental and nutrient signals [4, 5]. It can grow either in the form of metabolically active vegetative cells or can initiate the sporulation process resulting in the formation of metabolically inactive bacterial spores [6, 7]. *B. anthracis* spores are the primary infectious form of anthrax that are synthesized under a variety of stress/unfavorable conditions such as nutrition deprivation. Once the bacteria encounter favorable conditions, these spores germinate into the vegetative form resulting in the clinical manifestation of anthrax disease [8, 9]. The virulence of *B. anthracis* is principally determined by three key factors—anthrax toxins (edema and lethal), a weakly immunogenic poly-γ-D-glutamic acid capsule, and bacteremia-associated host tissue damage [10–13]. Anthrax toxins and capsule-synthesizing proteins are encoded by two separate extrachromosomal plasmids—pXO1 (182 kbp) and pXO2 (96 kbp), respectively. Curing of either plasmid greatly reduces virulence [8, 14, 15]. The anthrax toxin genes–*pagA* [encoding protective antigen (PA)], *cya* [encoding edema factor (EF)] and *lef* [encoding lethal factor (LF)] encoded by pXO1 plasmid are required for the functional synthesis of bipartite edema toxin (PA and EF) and lethal toxin (PA and LF) [10, 16–18]. The expression of these genes is maximal during exponential growth, implicating growth-phase dependent regulation of toxin synthesis during infection [19–21]. Furthermore, pXO1 also encodes for the global virulence gene regulator anthrax toxin activator (AtxA) protein that regulates the expression of anthrax toxin genes as well as capsule activator genes [22–25]. A high level of AtxA protein is imperative to produce anthrax toxins, and strains lacking AtxA show little to no anthrax toxin production [26, 27]. AtxA expression level is further antagonistically regulated by the global transcriptional regulators AbrB and CodY that are shown to interact with different regions of *atxA* gene promoter [20, 28–30].

The transition from one growth phase to another in a bacterial life cycle is a highly orchestrated mechanism involving various regulatory proteins and post-translational modifications such as protein phosphorylation [31–34]. Protein phosphorylation in prokaryotes was first demonstrated by Garnak and Reeves in 1979 as involving the His/Asp kinase enzymes of two-component regulatory systems [35]. Phosphorylation on serine and threonine residues had previously been identified in eukaryotes in the mid-1900s [36, 37], and for a long time this regulation was thought to be missing in prokaryotes.

In *B. anthracis*, while ser/thr protein phosphorylation is implicated in various cellular pathways including bacterial pathogenesis, only a few ser/thr kinases (STK) and ser/thr phosphatases (STP) have been characterized [38–46]. These include the three STKs (PrkC, PrkD, and PrkG) and a single STP (PrpC). In the present study, we characterize a second ser/thr phosphatase–PrpN (GBAA_RS03150/ BAS0539), annotated as a putative STP in the NCBI database. The functional relevance of PrpN in *B. anthracis* physiology is examined using a null mutant

strain of *prpN*. Our results identify CodY, a pleiotropic global transcription regulator, as a substrate of serine/threonine phosphatase PrpN and serine/threonine kinase PrkC in *B. anthracis*. Most importantly, this study demonstrates CodY phosphorylation to be a regulatory switch in the synthesis of anthrax toxins.

## Results

### Identification and characterization of PrpN protein

BAS0539 was identified as a putative *B. anthracis* ser/thr phosphatase in the NCBI database's annotation. Conserved domain analysis of the retrieved protein sequence using CDD tool (NCBI) confirmed the presence of a metal-dependent protein phosphatase (MPP) domain characteristic of a large family of mostly eukaryotic-like ser/thr phosphatases (cd00144: MPP_PPP_family), with all the conserved residues of the active and metal binding sites (Fig 1A). We chose to designate this putative phosphatase as PrpN. The PrpN-encoding gene is flanked in the *B. anthracis* chromosome by a glycerol-3-phosphate ABC transporter/glycerol-3-phosphate-binding protein and a DNA-binding response regulator (Fig 1B). The DOOR and MicrobesOnline databases suggested that these genes are not a part of an operon. This prediction was validated using intergenic primers as depicted in the schematic to assess co-transcription of the BAS0538, BAS0539 and BAS0540 genes by RT-PCR (Fig 1B). A reaction with no reverse transcriptase (NRT) was used as a negative control to check the presence of contaminating genomic DNA in the mRNA sample, while genomic DNA template was used as a positive control. Amplified products of around 1200 bp were observed in lanes 2 and 4 respectively (genomic DNA as template), while no amplification was observed in lane 3 and 5 (NRT) confirming independent transcription of these genes (Fig 1C—left panel). Additionally, *rpoB* and *prpN* intergenic primers amplifying a product of around 100-bp were used to ensure successful cDNA synthesis (Fig 1C—right panel). Protein BLAST analysis showed 99.1% sequence identity of PrpN with putative phosphatases in many strains of *Bacillus cereus* and *Bacillus thuringiensis* and 95.3% sequence identity with *Bacillus subtilis* strains (S1A Fig). Evolutionary relationship of PrpN among firmicutes was also examined by generation of a phylogenetic tree (S2 Fig). Interestingly, the observed hits were also annotated as STPs. Also, *Streptococcus pneumoniae*, a non-sporulating pathogen was clustered in the clade comprising sporulating Bacillus strains. Structural prediction using I-TASSER suggested structural similarity to previously characterized ser/thr phosphatases namely, those of *Gallus gallus* (PDB ID- 1S70) and *Escherichia phage lambda* (PDB ID- 1G5B) and the presence of a manganese ion ($Mn^{+2}$) binding pocket (S1B and S1C Fig).

To biochemically characterize PrpN we cloned, expressed, and purified the protein (as described in Materials and Methods section). We then used the non-specific phosphatase substrate p-nitrophenyl phosphate (pNPP) to confirm enzyme activity and to optimize assay conditions and analyze co-factor requirements. The lone characterized *B. anthracis* ser/thr phosphatase, PrpC, was used as a positive control and EDTA/EGTA control was used to examine metal ion requirements. The phosphatase assay data indicated $Mn^{+2}$ as the most preferred metal ion while other ions (zinc, magnesium and calcium) were not able to substitute for $Mn^{+2}$ in PrpN activation [Fig 1D-(i)]. Following this, we then used a more specific ser/thr phosphatase assay kit containing ser/thr phosphopeptides and checked for PrpN phosphatase activity in the presence and absence of ser/thr phosphatase inhibitors. The readings were plotted using GraphPad PRISM software and enzyme kinetics parameters for PrpN were calculated using the Michaelis Menten equation [Fig 1D-(ii)]. The above experimental data confirmed PrpN as the second functional ser/thr phosphatase in *B. anthracis*.

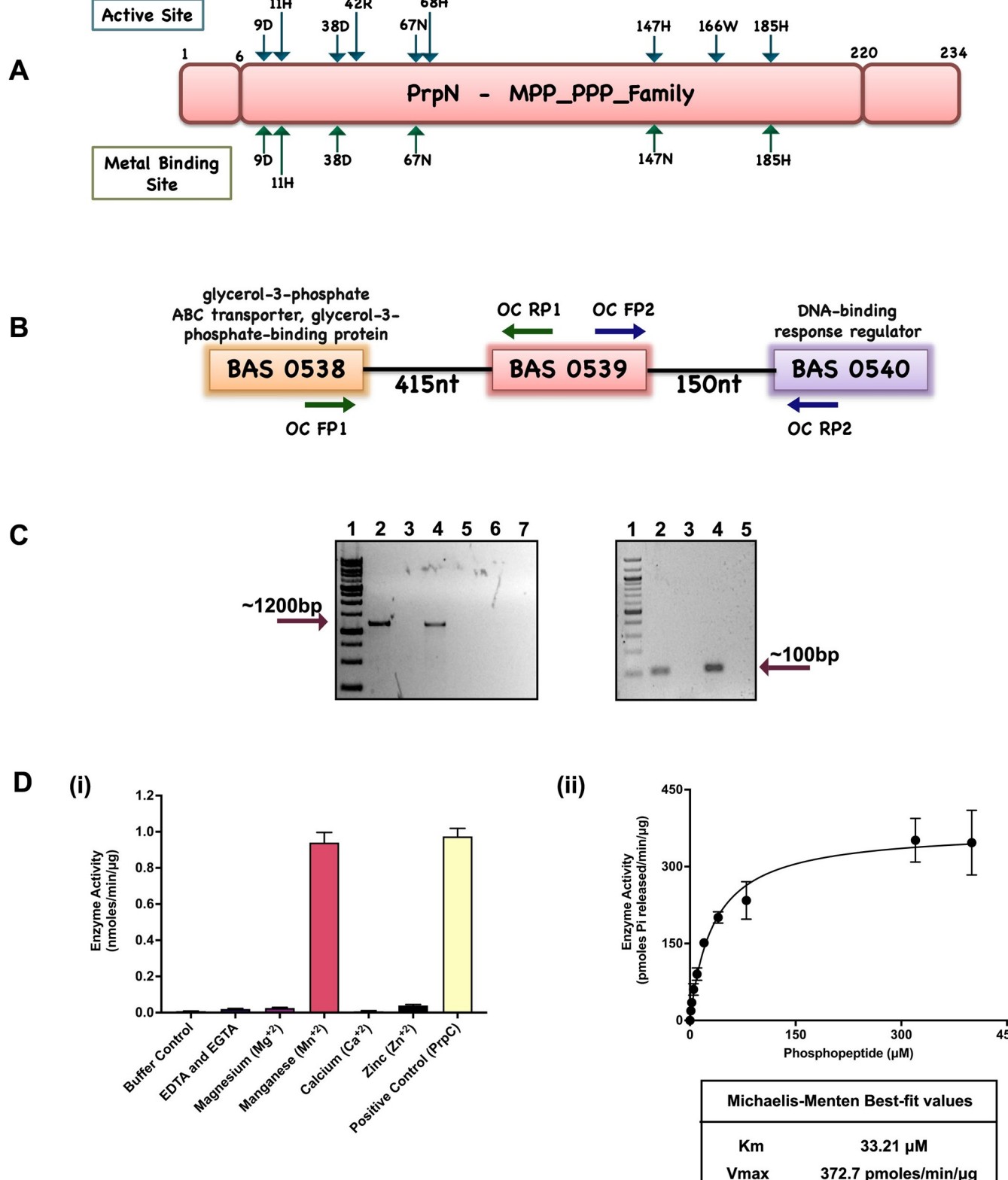

**Fig 1. Genomic organization and biochemical characterization of PrpN.** (A) Schematic description of PrpN phosphatase domain. PrpN primary sequence analysis depicted the presence of metallophosphatase (MPP) phosphoprotein phosphatase (PPP) conserved domain family. The conserved active site and metal binding site residues are illustrated in the schematic (B) Pictorial representation of *prpN* gene organization in *B. anthracis* genome. The reference organization

was fetched from NCBI GenBank database. (C) Operon prediction by RT-PCR analysis of BAS0538-40 region. BAS0538, 39 and 40 internal gene specific primers as depicted in Fig 1B were designed for PCR to amplify gene products of around 1200-bp. Agarose gel showing the amplified products by the primer pairs (OC Fp1 and OC Rp1) and (OC Fp2 and OC Rp2). Left panel- Lane 1: 1Kb DNA Ladder RTU (GeneDirex, Cat. No. DM010-R500), Lane 2 and 4: gDNA as template, Lane 3 and 5: cDNA as template, Lane 6 and 7 (No-RT cDNA as template). Right panel- Agarose gel showing successful cDNA preparation using *rpoB* and *prpN* intergenic primers amplifying a gene product of 104-bp and 120-bp, respectively. Lane 1: 100bp DNA Ladder H3 RTU (GeneDirex, Cat. No. SD003-R600), Lane 2: cDNA as template with *rpoB* primers, Lane 3: No-RT cDNA as template with *rpoB* primers, Lane 4: cDNA as template with *prpN* primers and Lane 5: No-RT cDNA as template with *prpN* primers. (D) Enzymatic and biochemical characterization of PrpN. (i) PrpN phosphatase activity was evaluated by pNPP hydrolysis assay in the presence of different ions to assess cofactor requirement. (ii) Kinetic plot of PrpN using serine/threonine phosphopeptides. The data were fitted to a Michaelis-Menten curve to determine enzyme kinetics parameters (Km and Vmax) using GraphPad Prism. Error bars reflect the variation of triplicate measurements.

## PrpN knockout and complemented strain generation

To study the function of the PrpN ser/thr protein phosphatase in the physiology of *B. anthracis*, a null mutant of *prpN* (BAS Δ*prpN*) was generated in *B. anthracis* Sterne strain background by a Cre-loxP strategy [47]. The complemented strain (BAS Δ*prpN*::*prpN*) was generated by electroporating the BAS Δ*prpN* strain with a pYS5 shuttle vector carrying the *prpN* gene along with its predicted promoter sequence (400-bp upstream region) (S3A Fig). Genomic DNAs of BAS WT, BAS Δ*prpN* and BAS Δ*prpN*::*prpN* were used as templates along with *prpN* gene specific primers to confirm these strains at the genomic level (S1 Table). An amplified product corresponding to *prpN* gene size (705 bp) was detected in BAS WT and BAS Δ*prpN*::*prpN*, while no amplification product was detected in BAS Δ*prpN* (S3B Fig). Next, the strains were confirmed at the protein level using PrpN specific antibody and purified recombinant PrpN protein as a positive control. A band consistent with the molecular weight of monomeric PrpN protein (~26 kDa) was detected in BAS WT and BAS Δ*prpN*::*prpN* lysates and no band was detected in the null mutant strain, confirming the deletion of the *prpN* gene (S3C Fig).

## Temporal expression of PrpN and its role in growth cycle

The role of protein ser/thr phosphorylation in bacterial growth and cell division is a widely investigated topic [48–50]. A recent report by our group in *B. anthracis* showed growth defects in a ser/thr kinase (PrkC) mutant strain, thus establishing a functional correlation between bacterial growth and ser/thr phosphorylation [41]. To examine the role of PrpN in *B. anthracis* growth machinery, we monitored the growth patterns of BAS WT, BAS Δ*prpN* and BAS Δ*prpN*::*prpN* strains. The bacterial cells were periodically examined microscopically to visualize the vegetative cells morphology. As seen in Fig 2A, BAS Δ*prpN* strain entered the stationary phase sooner and showed reduced growth yield as compared to BAS WT strain, while the growth pattern of BAS Δ*prpN*::*prpN* strain was comparable to BAS WT strain. We also monitored the growth profiles in the presence of salt [1 M sodium chloride (NaCl)] and oxidative [2.5 mM hydrogen peroxide ($H_2O_2$)] stresses and observed an extended lag phase in the absence of *prpN* (Fig 2B and 2C). Intrigued by the above growth pattern, we next checked for the temporal expression of PrpN in BAS WT by immunoblot analysis at four bacterial growth phases–(i) lag, (ii) exponential, (iii) early stationary (time points indicated in Fig 2A) and (iv) late stationary/sporulation initiation ~60 hours [41]. The bacterial cells were visualized microscopically to confirm the growth phase (Fig 2D- upper panel) prior to protein lysates preparation. Bacterial cells of BAS Δ*prpN* strain were also examined at the above-mentioned time points to compare the bacterial morphology with BAS WT. The deletion of *prpN* resulted in twisted and coiled-coil chains and this phenotype was most prominent at stationary time points (Fig 2D- lower panel). GroEL was used as a loading control for densitometer analysis and the results demonstrated maximal expression of PrpN in the stationary growth phase (Fig 2E and 2F). Interestingly, PrpN showed two bands at stationary growth phase in the

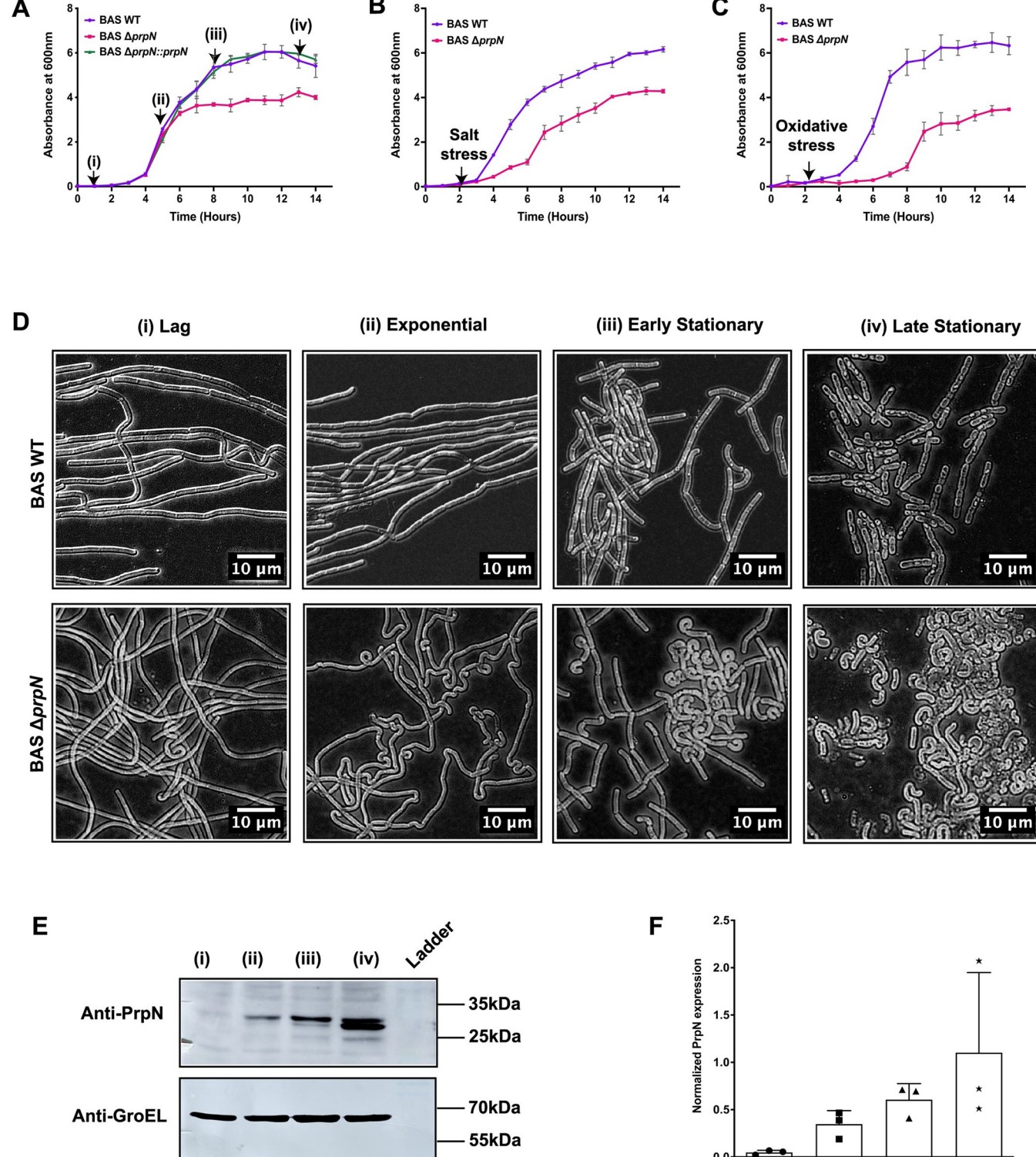

**Fig 2. PrpN is critical for optimum vegetative growth and is maximally expressed during stationary growth phase.** Growth patterns of indicated strains at 37˚C monitored by measuring $A_{600nm}$ at intervals of 2 hours till 14 hours. Average values and standard deviation calculated from biological triplicates are shown in the graph. Triplicate cultures were grown in LB medium to 0.1 $A_{600nm}$ and then stresses imposed by adding (A) no addition, (B) 1 M NaCl, and (C) 2.5 mM $H_2O_2$. (D) Representative phase contrast microscopy images of BAS WT and BAS Δ*prpN* strains at different time points. Scale bars are depicted in the images. (E) Expression of PrpN at different growth phases. Representative immunoblot showing growth dependent differential expression of PrpN in BAS WT

strain. Equal amounts of protein lysates prepared from different growth phases (as indicated in Fig 2A and 2D) were loaded and probed using anti-PrpN and anti-GroEL. Ladder- PageRuler Prestained Protein Ladder, Thermo-Scientific (Cat. No. 26616). (F) Densitometer analyses were done using Amersham Imager600 software and the corresponding PrpN/GroEL ratios were plotted using GraphPad Prism. Densitometer readings calculated from three experiments executed independently are shown in the bar graph.

immunoblot. A possible explanation for this could be some post-translational modification of PrpN in late stationary growth phase. Further investigation is required to confirm this hypothesis. Overall, the above results indicate the possible role of PrpN in bacillus growth in normal as well as stress conditions and a growth phase dependent temporal expression of PrpN in *B. anthracis*.

## Absence of PrpN causes structural abnormalities in vegetative bacterial cells

Morphological variations and multi-cellular arrangements such as bacterial chaining, biofilm formation, and host colonization are often linked to bacterial survival and virulence in several pathogenic bacteria including *B. anthracis* [41, 51–54]. Bacillus are rod-shape chain-forming bacteria that grows along their longitudinal axes facilitated primarily by re-arrangements of cell wall and cytoskeleton proteins, thereby maintaining a homogenous population of bacterial cells in normal growth conditions [55, 56]. Since our previous results implicated PrpN in *B. anthracis* growth control, we examined BAS WT, BAS Δ*prpN* and BAS Δ*prpN*::*prpN* cells at exponential and stationary phase by phase contrast microscopy (Fig 3A). A heterogeneous population of bacterial cells and structural aberrations such as coiled-coil structures and bent cells were observed in the BAS Δ*prpN* strain. Interestingly, these variations were more prominent at stationary growth phase synchronizing with PrpN expression level (Fig 3A—lower middle panel). We then visualized these cells at ultra-structural level by scanning electron microscopy to get a better insight of this phenotype. Strains lacking *prpN* showed structural deformities and coiled vegetative bacilli mostly at the chain ends and septal regions, while the complemented strain bacilli were similar to BAS WT strain (Fig 3B). Since, accurate cell septation is crucial for bacterial growth and cell division, we also looked for septation pattern using FM4-64 membrane staining dye in an agarose pad setup (details in material and methods section). Confocal micrographs showed a high population of multi-septa cells (18.2%) in the BAS Δ*prpN* strain (Fig 3C- middle panel and 3D), while BAS WT and BAS Δ*prpN*::*prpN* strain showed an evenly distributed septation pattern (Fig 3C—left and right panel). These septation abnormalities could be a possible reason for the loss of homogeneity in strains lacking *prpN* since the null mutant strain showed bacillus cells with varying lengths and a high coefficient of variation (S4 Fig). Interestingly, morphological variations and a similar multi-septation phenotype were reported by our group in the absence of the ser/thr kinase PrkC in *B. anthracis* [41]. This indicates that ser/thr phosphorylation is pivotal for normal vegetative bacterial cell morphology and septation in *B. anthracis*.

## PrpN is necessary for sporulation process

*B. anthracis* spores are the infectious form of anthrax that can infect the host via three different routes: inhalation, gastrointestinal or cutaneous. These anthrax spores were even used as potential biowarfare agents posing a serious threat to not just cattle but even humans [1, 57, 58]. Thus, understanding the process of sporulation is crucial in the context of anthrax pathogenesis. While the role of bacterial STKs and STPs in *B. subtilis* sporulation is an extensively researched and reviewed topic [59–63], there are few studies done in *B. anthracis*. Our results above indicated maximal expression of PrpN protein in stationary growth phase suggesting its

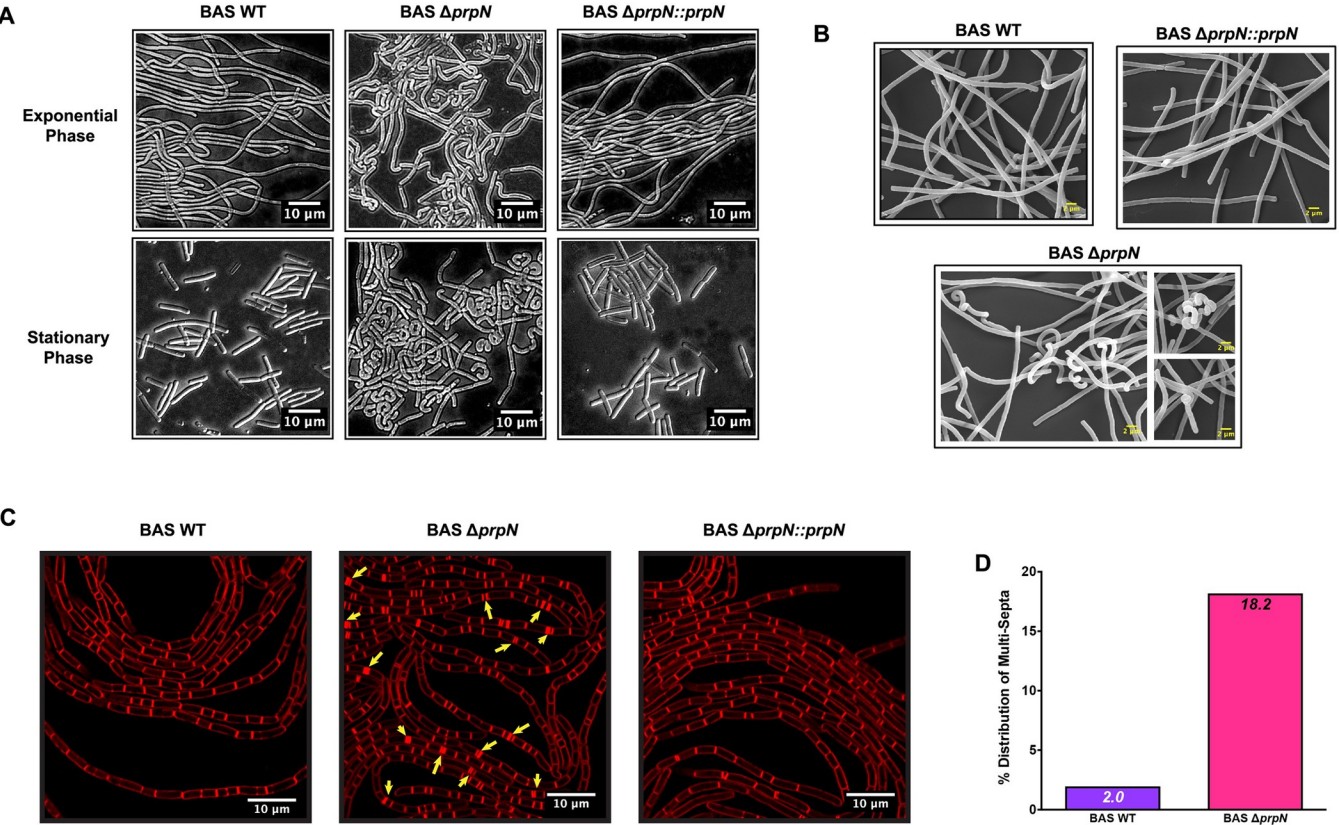

**Fig 3. Effect of *prpN* deletion on vegetative cell morphology.** (A) Representative phase contrast images of bacterial cells at exponential and stationary growth phase. Bacterial cells were visualized under 100x/1.4 oil DIC objective of Zeiss Axio Imager Z2 upright microscope. Scale bars represents 10 μm. (B) Representative scanning electron microscope images of indicated vegetative bacterial cells. Cells were visualized under Zeiss Evo LS15. Scale bars represent 2 μm, magnification-5000X. (C) Cell septation properties of BAS WT, BAS Δ*prpN* and BAS Δ*prpN::prpN*. Bacterial strains grown in an LB agarose pad setup for 6 hours containing 1 μg/mL FM4-64 membrane staining dye. Live vegetative bacterial cells were visualized using Leica SP8 confocal laser scanning microscope. Arrow heads indicate multi-septation in the BAS Δ*prpN* strain (middle panel). Scale bars represents 10 μm. (D) Graph indicating % distribution of multi-septa in 1500 BAS WT and BAS Δ*prpN* cells.

possible role during the sporulation process. To examine this hypothesis, we first microscopically monitored BAS WT and BAS Δ*prpN* bacterial cells in agarose pad prepared with sporulation media. Images were captured at five time points– 24, 48, 72, 96 and 144 hours (Fig 4A). BAS WT showed initiation of sporulation process at 72 hours and release of matured spores at 96 hours (Fig 4A—above panel), while in the BAS Δ*prpN* strain severe defects in sporulation process with multiple coiled-coil vegetative bacilli were observed until 96 hours (Fig 4A—below panel). To quantitate this result, sporulation efficiencies and total spore counts of both the strains along with the complemented strain were determined. A severe sporulation defect in the absence of *prpN* was confirmed while this defect was rescued in the complemented strain (Fig 4B and 4C). These results confirmed a requirement for PrpN in *B. anthracis* sporulation.

Sporulation is a highly regulated process that involves asymmetric septation of the vegetative cells resulting in the generation of two distinct compartments—mother cell and forespore, forespore engulfment by mother cell, spore maturation and finally the release of mature spore following mother cell lysis [64–66]. As sporulation defects were evident in our null mutant strain, we examined BAS WT, BAS Δ*prpN* and BAS Δ*prpN::prpN* strains grown in sporulation medium at the ultrastructural level by transmission electron microscopy (TEM). To visualize

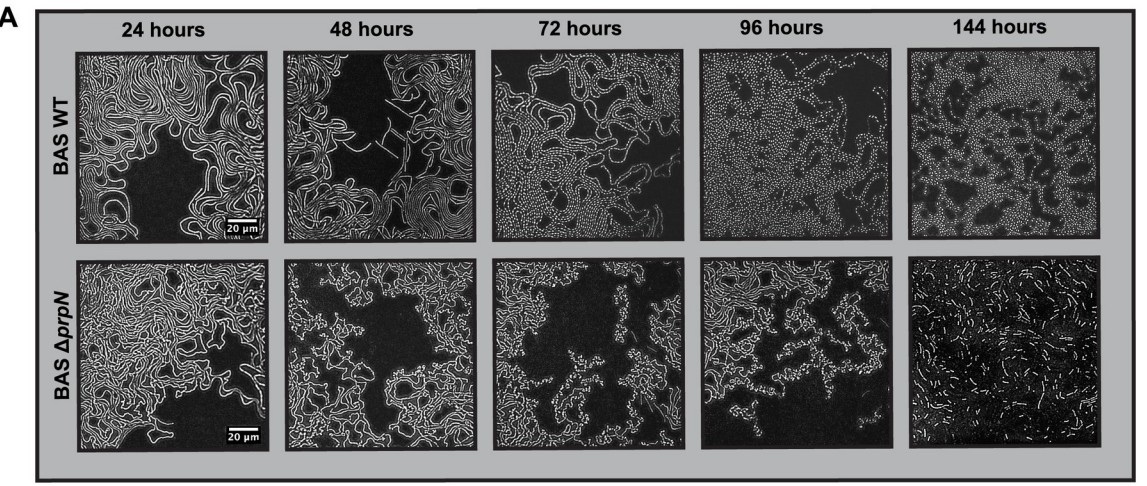

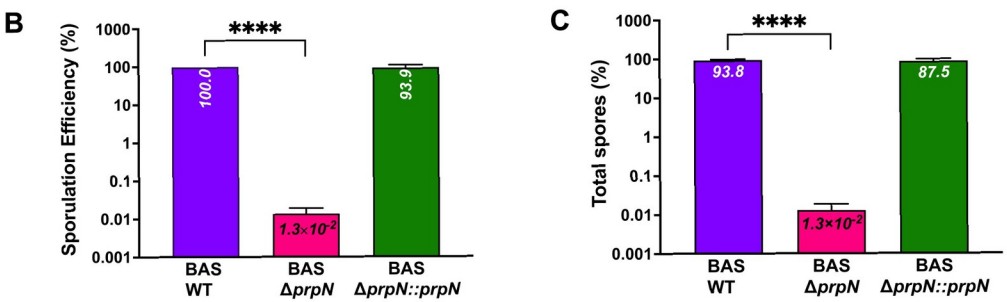

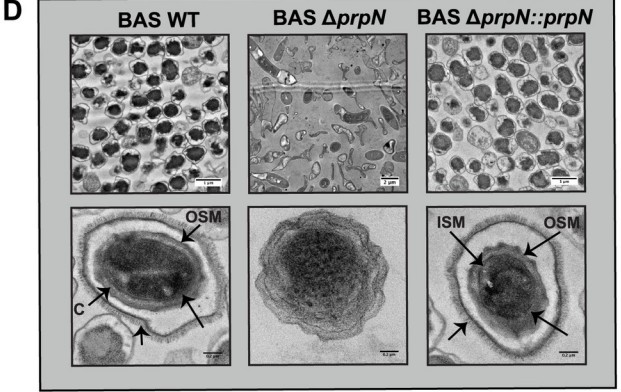

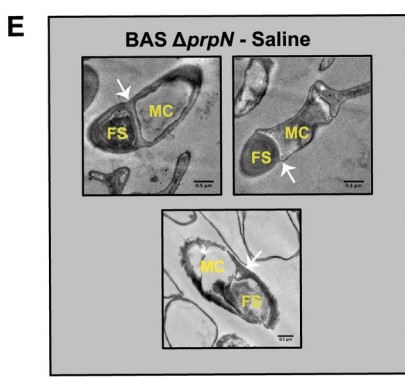

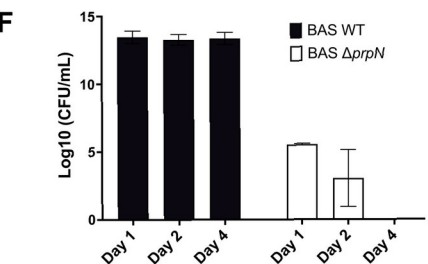

**Fig 4. Effect of *prpN* deletion on sporulation.** (A) Representative phase contrast images of BAS WT and BAS Δ*prpN* strains grown in agarose pad prepared in sporulation medium. Images were captured at the indicated time points as depicted in the Fig panel using Zeiss Axio Imager Z2 upright microscope. Scale bars represents 20 μm. (B) and (C) Bar graphs depicting sporulation efficiency and total spores in indicated strains. Average values and standard deviations calculated from three experiments executed independently are shown in the bar graphs. Asterisks indicate statistical significance of the data set calculated using two-tailed Student's *t* test. **** denotes $p < 0.0001$. (D) Representative transmission electron micrographs of indicated strains spores. The strains were grown in sporulation medium for 72 hours and washed with water to remove vegetative cell debris. The spore pellet was processed for TEM imaging and visualized using FEI Tecnai G2 Spirit at 200 KV. Spore layers are depicted in the images with arrows- ES: exosporium, OSM: outer spore membrane, ISM: inner spore membrane, C: cortex. Scale bars are depicted in the images. Magnification is 550X (upper panel) and 15000X (lower panel). (E) Representative transmission electron micrographs of BAS Δ*prpN* sporulating cells. FS denotes forespore and MC denotes mother cell. White arrow indicates asymmetric septation. Images were captured using FEI Tecnai G2 Spirit at 200 KV. Scale bars represents 0.5 μm. (F) Bar graph depicting viable cell count in BAS WT and BAS Δ*prpN* post complete sporulation. Data is represented as mean CFU $\log_{10}$/mL. Error bars denote standard deviations of three independent experiments.

intact cells in the sporulation medium, the samples were prepared by two different methods (details in material and methods section). BAS WT and BAS Δ*prpN*::*prpN* spore images showed intact spores having multiple protective layers (Fig 4D—left and right panel), while undeveloped spores and vegetative cells with asymmetric septation were seen in the BAS Δ*prpN* strain (Fig 4D—middle panel and 4E). Following this, we next wanted to analyze the ultimate fate of the vegetative cell population and undeveloped spores post sporulation event. For this, we examined the viability of the BAS Δ*prpN* strain for four days after complete sporulation was observed in BAS WT strain. Bacterial cells were harvested and CFU were determined without heat treatment to detect both vegetative bacilli and any viable spores (See Material and Methods). Absence of colonies post 2–4 days confirmed the death of vegetative bacilli and also ruled out the possibility of delayed sporulation in the BAS Δ*prpN* strain (Fig 4F). These results signify the indispensable role of PrpN during sporulation process in *B. anthracis*.

## PrpN is required for efficient toxin production

Since, the expression of anthrax exotoxins is crucial for the pathogenic cycle of *B. anthracis*, we next examined the role of PrpN in toxin synthesis as well as secretion. For this, BAS WT, BAS Δ*prpN* and BAS Δ*prpN*::*prpN* strains were grown in NBY medium containing 1% $NaHCO_3$ for toxin proteins synthesis [67–69]. Whole cell lysates and protein obtained from culture supernatant fractions were probed using PA and LF specific antibodies to examine the synthesis and secretion of toxin proteins. Whole cell lysates immunoblot were stripped and probed again using GroEL antibody for normalization. A drastic decrease in both toxin synthesis (~80% reduction—PA and LF) and secretion (~75% reduction—PA and ~70% reduction—LF) was evident in the BAS Δ*prpN* strain, while an episomal copy of *prpN* in the mutant strain background (BAS Δ*prpN*::*prpN*) was able to rescue this defect and toxin production was comparable to BAS WT (Fig 5A, 5B, 5C and 5D). These results clearly indicate the vital role of PrpN in *B. anthracis* toxin proteins expression.

Since, defective PA and LF expression was evident in the *prpN* null mutant strain, we checked for the expression of anthrax toxin activator protein—AtxA (master regulator of anthrax virulence), a prerequisite for anthrax toxins and capsule protein synthesis [20, 22, 27]. More recently, it was reported that AtxA binds specifically to the promoter region of *pagA* (encoding PA) and thereby positively regulates its expression [24]. Immunoblot analysis using AtxA specific antibody showed a significant decrease (~60% reduction) in AtxA protein expression in the BAS Δ*prpN* strain with respect to control strain (BAS WT) (Fig 5E). The complemented strain (BAS Δ*prpN*::*prpN*) showed an AtxA protein level similar to that in the BAS WT strain (Fig 5E), confirming the role of PrpN in regulation of AtxA protein level.

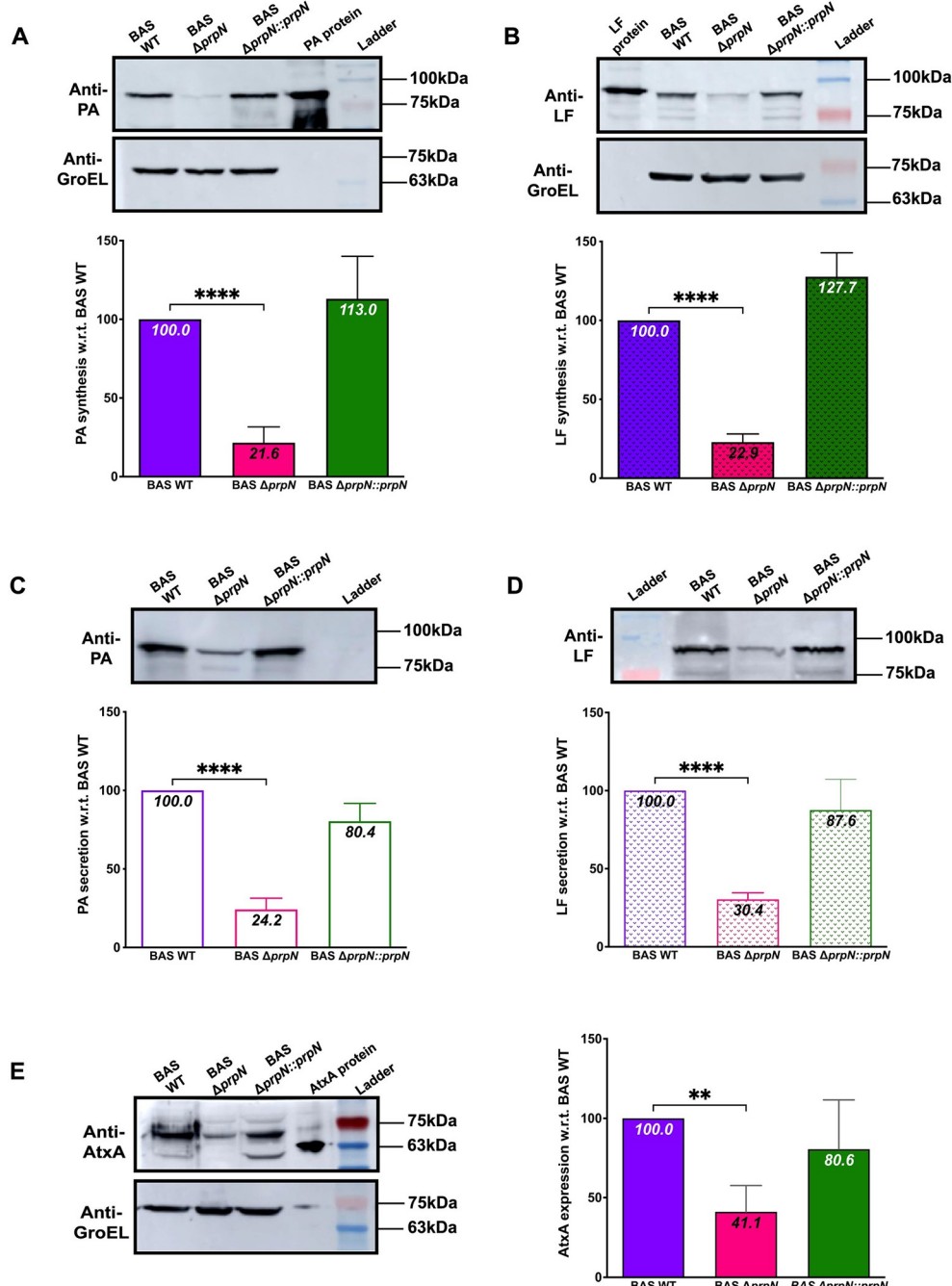

**Fig 5. Defective PA, LF and AtxA expression in BAS Δ*prpN* strain.** Representative immunoblots and bar graphs showing PA, LF, and AtxA synthesis in indicated strains. The strains were grown in NBY medium containing 1% NaHCO₃. Whole cell lysates (A,B,E) or supernates (C,D) were loaded in equal amounts and probed using anti-PA (A, C), anti-LF (B,D), or anti-AtxA (E). The blots were stripped and probed using anti-GroEL. Densitometer analyses were done using Amersham Imager600 or ImageLab software and the corresponding PA, LF, and AtxA ratios to GroEL were plotted using GraphPad Prism. Average values and standard deviations calculated from minimum three independent experiments are shown in the bar graphs. Ladder- BlueRAY Prestained Protein Ladder, GeneDirex (Cat. No. PM006-0500). Statistical Analysis: Asterisks indicate statistical significance of the data set calculated using two-tailed Student's *t* test. * corresponds to p<0.05; ** corresponds to p<0.01; *** corresponds to p<0.001 and **** corresponds to p<0.0001.

Taken together, these results suggest that PrpN positively regulates toxin synthesis in *B. anthracis* at least in part by maintaining the normal amount of AtxA.

## CodY is a target of PrpN and PrkC

The data presented so far demonstrates the important role of PrpN in several life stages of *B. anthracis*, but the underlying PrpN-mediated dephosphorylation events remained elusive. To examine this question, CodY, a global transcriptional regulator reported to be important for growth, sporulation and toxin synthesis in *B. anthracis* was selected for our study [30, 70–73]. In *B. subtilis*, CodY phosphorylation was detected at a serine residue (Ser215) located in its highly conserved DNA-binding domain (helix-turn-helix motif) [74–76]. To examine CodY as a potential PrpN target, we cloned the *codY* gene with a hexa-histidine tag in pYS5 shuttle vector and electroporated the resulting over-expression plasmid in BAS WT and BAS Δ*prpN* strains. The recombinant CodY protein was purified from both the strains by affinity purification and resolved on SDS-PAGE (Fig 6A). Next, the relative phosphorylation status of CodY protein was estimated by anti-phosphoserine antibody after normalization to total protein amount. Densitometer analysis showed that CodY from BAS Δ*prpN* strain contained 3-fold more phosphate than CodY from BAS WT strain (Fig 6B and 6C). Additionally, PrpN mediated dephosphorylation of CodY protein purified from the BAS Δ*prpN* strain was estimated by measuring the generated inorganic phosphate in the reaction, thus confirming CodY as a target of PrpN (Fig 6D).

Since, protein phosphorylation at ser/thr residues is regulated by ser/thr kinases and ser/thr phosphatases, we next explored ser/thr kinase mediated phosphorylation of CodY. In view of this, an *in-vitro* kinase assay was performed using the catalytic (cat) domain of *B. anthracis* ser/thr kinase, PrkC (GST tag) and CodY (His$_6$ tag). CodY alone and CodY incubated with ATP were used as controls and the phosphorylation signal was detected using anti-phospho-serine antibody and Pro-Q diamond stain. No signal was observed in lane 1 (CodY) and lane 2 (CodY + ATP), while positive signals were observed in lane 4 corresponding to PrkC (autophosphorylation) and CodY (Figs 6E and S5A). Additionally, we probed CodY (His$_6$ tag) produced in *E. coli* from pETDuet vector system in the absence (lane 1) and presence of *prkCcat* (lane 2) encoding gene. CodY (His$_6$ tag) produced in *E. coli* from pProEXHTc (lane 3 and 4) was used as a negative control. CodY produced in the presence of PrkC showed a positive signal, while no signal was observed in the absence of PrkC (Fig 6F). These results confirmed PrkC mediated phosphorylation of CodY *in-vitro*. Next, to examine the *in-vivo* phosphorylation status of CodY via PrkC, we overexpressed CodY (His$_6$ tag) in BAS WT and BAS Δ*prkC* strains and probed the purified CodY proteins using anti-phosphoserine. CodY produced in the absence of PrkC showed no difference in its phosphorylation level when compared to BAS WT (S5B Fig). A possible reason for this difference *in-vivo* could be regulation of CodY phosphorylation by other STKs such as PrkD and PrkG, thereby compensating for the loss of PrkC. Overall, these results confirmed CodY as a target of serine/threonine kinase PrkC and serine/threonine phosphatase PrpN in *B. anthracis*.

## Phosphorylation-mediated regulation of CodY activity (at serine215) by PrpN

CodY is a pleiotropic transcriptional regulator that modulates the expression of various genes at exponential as well as stationary growth phase and shows an unaltered expression profile throughout the bacterial growth [72] (S6 Fig). The logical next step was to identify effects resulting from PrpN-mediated CodY dephosphorylation. Since, the serine215 phosphorylation site in CodY protein is located in its DNA binding region, we suspected this modification

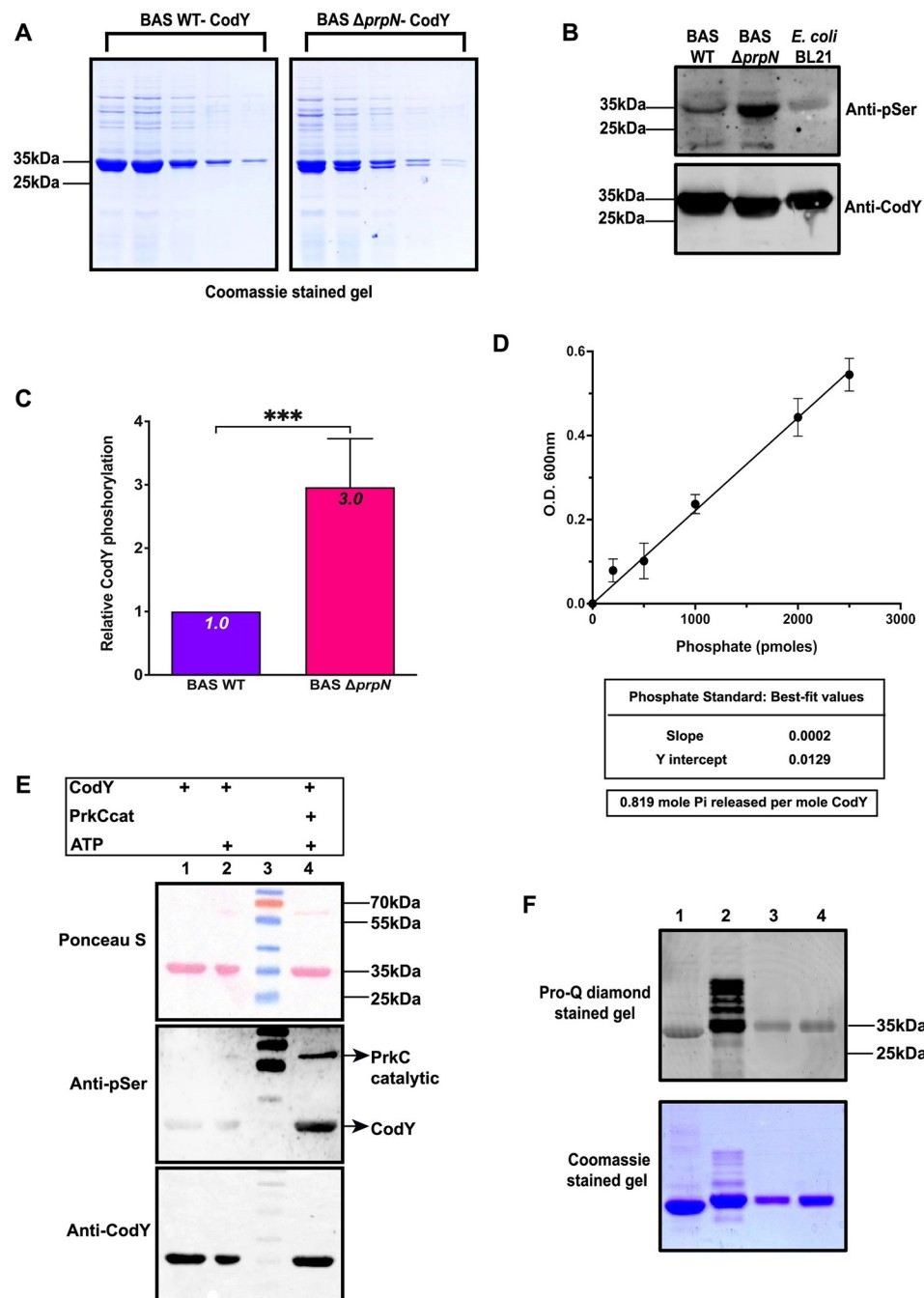

**Fig 6. CodY—A substrate of PrkC and PrpN.** (A) Coomassie stained gel image of CodY elution's from BAS WT and BAS Δ*prpN*. (B) and (C) Representative immunoblots and bar graph showing phosphorylation status of CodY protein purified from indicated strains. Equal amounts (1 μg) of purified CodY proteins were loaded and probed using anti-phosphoserine. The same blot was stripped and probed again using anti-CodY. Indicated MWs were derived from adjacent lanes containing PageRuler Prestained Protein Ladder, Thermo Scientific (Cat. No. 26616). Densitometer analysis were done using Amersham Imager600 software and the corresponding phosphoserine/CodY ratio with respect to BAS WT were plotted in a bar graph using GraphPad Prism. Average values and standard deviations calculated from five independent experiments are shown in the bar graph. Asterisks indicate statistical significance of the data set calculated using two-tailed Student's *t* test. *** corresponds to p<0.001. (D) Phosphate standard curve and results for amounts of inorganic phosphate released from phosphorylated CodY proteins purified from BAS Δ*prpN* strain by PrpN treatment. (E and F) PrkC mediated phosphorylation of CodY. (E) Representative immunoblots and Ponceau-S stained image of *in-vitro* kinase assay performed using autophosphorylated PrkCcat and CodY. (F) Co-expression of CodY-His$_6$ with and without PrkCcat. Proteins were resolved on SDS-PAGE and visualized using Pro-Q

diamond phospho specific gel stain to examine the phosphorylation status (upper image) and Coomassie stain to confirm loading pattern (lower image). Lane 1: CodY-His$_6$ produced in *E. coli* using pETDuet without PrkCcat; Lane 2: CodY-His$_6$ produced using pETDuet with PrkCcat; Lane 3 and 4: CodY-His$_6$ produced using pProEXHTc. Indicated MWs were derived from adjacent lanes containing PageRuler Prestained Protein Ladder, Thermo Scientific (Cat. No. 26616).

could modulate its DNA-binding affinity to its target genes [75]. To test this conjecture, we generated a CodY phosphomimetic protein by mutating serine215 to aspartate residue (CodYS215E) and a CodY phosphoablative by mutating serine215 to alanine residue (CodYS215A) (S7A Fig). The recombinant proteins were purified by affinity chromatography and the secondary structures were examined by circular dichroism (S7B Fig). Superimposed spectra of native CodY, CodYS215A and CodYS215E showed similar patterns, thus negating the possibility of mutation induced misfolding of CodY proteins (S7B Fig—Right panel). Additionally, to validate *in-vivo* phosphorylation of CodY at serine215 residue, we generated CodY phosphoablative (CodYS215A with His$_6$ tag) overexpression strains using pYS5 shuttle vector in the background of BAS WT (BAS WT::*codYS215AHis$_6$*) and BAS Δ*prpN* strains (BAS Δ*prpN*::*codYS215AHis$_6$*). CodYS215A-His$_6$ purified from these strains was probed using anti-phosphoserine. Auto-phosphorylating catalytic domain of PrkC (GST-tag) and CodY (His$_6$-tag) purified from *E. coli* BL21(DE3) strain were used as positive and negative controls, respectively. The immunoblot probed using anti-phosphoserine antibody showed a band only in lane 4 (positive control), while no phosphorylation was detected in lanes 2 and 3 (CodY-S215A-His$_6$ produced in bacillus strains) and lane 5 (negative control) confirming CodY phosphorylation at this specific residue (Fig 7A). Additionally, the mobility pattern of native CodY and CodYS215A produced in BAS WT and BAS Δ*prpN* strains was examined using Phos-tag gel system for separation of phosphorylated and unphosphorylated forms of CodY (Fig 7B—upper image). These proteins were also resolved on SDS-PAGE to compare the mobility pattern in the absence of Phos-tag (Fig 7B—lower image). Two prominent bands were observed in CodY-His$_6$ proteins produced in BAS WT (lane 2) and BAS Δ*prpN* (lane 3), while a single band was detected in CodYS215A produced in BAS WT (lane 1) and BAS Δ*prpN* (4) indicating serine215 as the lone phosphosite in CodY. Also, the upper band corresponding to pCodY in lane 2 and 3 was significantly reduced in lane 2 (CodY produced in BAS WT) compared to lane 3 (CodY produced in BAS Δ*prpN*) (Fig 7B—upper image). The above results confirm PrpN mediated dephosphorylation of CodY specifically at serine215 residue.

The promoter region of AtxA, previously validated as a CodY target in *B. anthracis* and downregulated in our study, was selected for DNA binding assays [29]. In *B. anthracis*, CodY positively regulates the accumulation of AtxA protein and toxin proteins and is reported to be crucial for virulence in animal model of anthrax infection [30, 73]. Electrophoretic mobility shift assay (EMSA) was performed using an increasing concentration of the CodY proteins (CodYS215E, CodYS215A and native CodY) and a fixed amount of *atxA* promoter as probe. A prominent DNA shift was evident in the case of CodY phosphoablative mutant (CodYS215A) and native CodY, while CodYS215E mutant showed no DNA binding with *atxA* promoter region even at higher protein concentration (Fig 7C). Interestingly, CodYS215A showed higher binding affinity with *atxA* as compared to native CodY suggesting a charged based interaction of CodY protein with its target DNA sequences (Fig 7C). We also examined CodY binding pattern in the presence of GTP and BCAA (branched chain amino acids), previously reported as positive regulators of CodY binding with its the target genes including *atxA* [29, 77–79]. As expected, native CodY and CodYS215A binding affinity was significantly enhanced in the presence of these effector molecules, while CodYS215E still showed no DNA binding (S7C Fig). Next, since the expression of AtxA protein was significantly reduced in the BAS

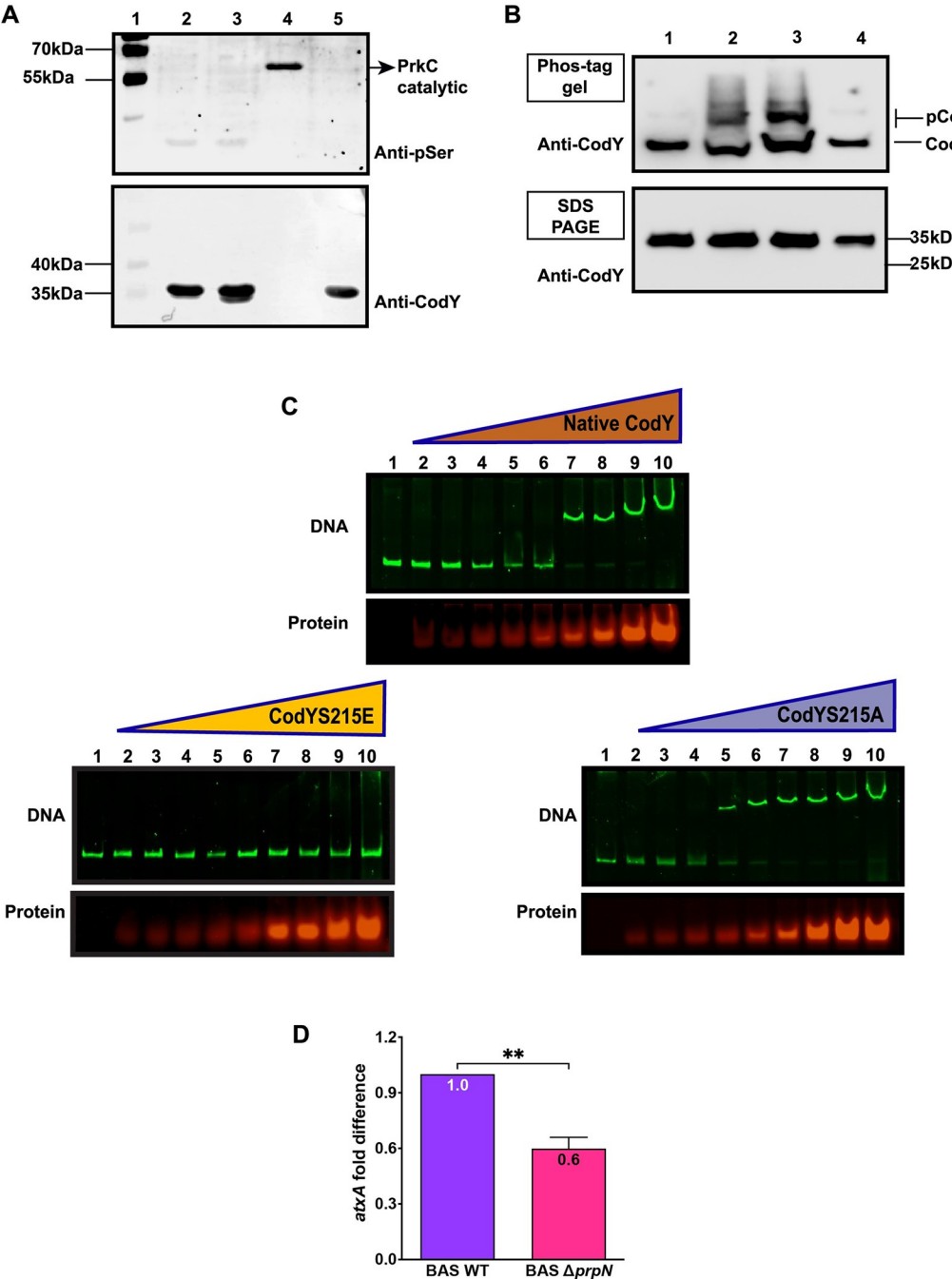

**Fig 7. *In-vivo* confirmation of CodY serine215 phosphosite in *B. anthracis* and impact of CodY phosphorylation on its DNA binding activity.** (A) Representative immunoblots to confirm *in-vivo* phosphorylation of CodY at serine215 residue. 2 µg CodYS215A-His$_6$ produced in BAS WT (lane 2) and BAS Δ*prpN* (lane 3) strains was resolved and probed using anti-phosphoserine. The same blot was stripped and probed using anti-CodY. 2 µg PrkCcat (lane 4) and native CodY (lane 5) produced in *E. coli* BL21 were used as positive and negative controls, respectively. Indicated MWs were derived from adjacent lanes containing PageRuler Prestained Protein Ladder (Lane 1), Thermo Scientific (Cat. No. 26616). (B) Representative immunoblots of native CodY and CodYS215A purified from bacillus strains resolved on Phos-tag precast gel (upper image) and SDS-PAGE (lower image). Lane 1: CodYS215A-His$_6$ produced in BAS WT, Lane 2: CodY-His$_6$ produced in BAS WT, Lane 3: CodY-His$_6$ produced in BAS Δ*prpN*, Lane 4: CodYS215A-His$_6$ produced in BAS Δ*prpN*. pCodY and CodY denotes phosphorylated and unphosphorylated form of CodY. Indicated MWs were derived from adjacent lanes containing PageRuler Prestained Protein Ladder, Thermo Scientific (Cat. No. 26616). (C) Electrophoretic Mobility-Shift Assay using SYBR Green and SYPRO Ruby stains. Increasing amounts (1, 1.5, 3, 4, 5, 6, 8, 10 and 12µM- lane 2 to 10) of native CodY, CodY S215E and CodY S215A proteins were added in a binding reaction containing 4 nM *atxA*

promoter region as probe. Lane 1 represents only DNA control. Upper gel image represents DNA bands stained using SYBR Green and lower image represents protein bands stained using SYPRO Ruby. (D) Comparative expression of *atxA* mRNA in BAS Δ*prpN* with respect to BAS WT strain. The RT-PCR data were normalized to the expression of *rpoB* from each strain. Error bars represents an average of three independent biological triplicates, each performed in three technical replicates.

Δ*prpN* strain (Fig 5E), we checked the transcript level of *atxA* mRNA in the BAS WT and BAS Δ*prpN* strains. Consistent with the AtxA protein levels, *atxA* mRNA expression was also downregulated in the BAS Δ*prpN* strain (Fig 7D). Finally, a CodY phosphoablative (CodYS215A) overexpression strain was generated in the BAS Δ*prpN* strain background (BAS Δ*prpN*::*codYS215A*) to mimic the *in-vivo* CodY phosphorylation level under native conditions (BAS WT). Toxins (PA and LF) and AtxA protein synthesis was examined and compared to BAS WT and BAS Δ*prpN* strains by immunoblot analysis (Fig 8A, 8B and 8C). The results indicated a partial reversion in toxins and AtxA protein level in the BAS Δ*prpN*::*codYS215A* strain, confirming unphosphorylated CodY as a positive regulator of AtxA and successive toxins synthesis in *B. anthracis*.

## Discussion

Reversible protein phosphorylation at specific ser/thr residues is an important post-translational modification by which physiological signals are transmitted and thereby regulate multiple cellular functions [80–83]. Ser/thr phosphorylation is catalyzed by ser/thr kinases by the transfer of phosphate group from ATP, while removal of phosphate group requires another class of enzyme namely, ser/thr protein phosphatases. Ser/thr protein phosphatases are further classified into three groups based on their protein structure and catalytic mechanism. These are metal-dependent protein phosphatases (PPM), phosphoprotein phosphatases (PPP) and haloacid dehalogenase (HAD). Amongst these, PPM is the most highly conserved STP family amongst all kingdoms of life [84]. Further, the role of STPs is extensively reviewed by several groups and linked to crucial cellular pathways such as cellular division, protein translation, immune response and pathogenesis in prokaryotes and eukaryotes [85–91]. For instance, *M. tuberculosis* encodes 11 STKs and a lone essential STP—PstP, which is reported to be vital for growth and virulence [92]. In *B. anthracis* too, three STKs (PrkC, PrkD and PrkG) and only one non-essential STP, PrpC have been functionally characterized [43, 45]. This was the driving thought of our study leading to the functional characterization of another ser/thr protein phosphatase—PrpN in *B. anthracis*. Sequence analysis of this putative ser/thr protein phosphatase revealed the presence of PPM family domain, also known as PP2C phosphatases (Fig 1A) [84]. Biochemical characterization of PrpN was done using synthetic ser/thr phosphopeptides to validate it as a ser/thr protein phosphatase (Fig 1D). The functional role of PrpN in *B. anthracis* life cycle and physiology was further explored by generating a *prpN* null mutant strain by Cre-loxP strategy in *B. anthracis* Sterne (BAS) vaccine strain background. The resulting strain is referred as BAS Δ*prpN* in the study. A complemented strain (BAS Δ*prpN*::*prpN*) was also made to confirm the direct role of PrpN and negate the possibility of any polar mutation. Functional characterization of these strains indicated the role of PrpN in multiple key cellular pathways as mentioned below:

a. Growth and cellular division machinery: attenuated growth profile and multi-septation,

b. Stress: sensitive to salt and oxidative stress,

c. Vegetative cells and spores morphology: coiled-coil vegetative bacilli (more prominent in the stationary growth phase) and defective spore layers,

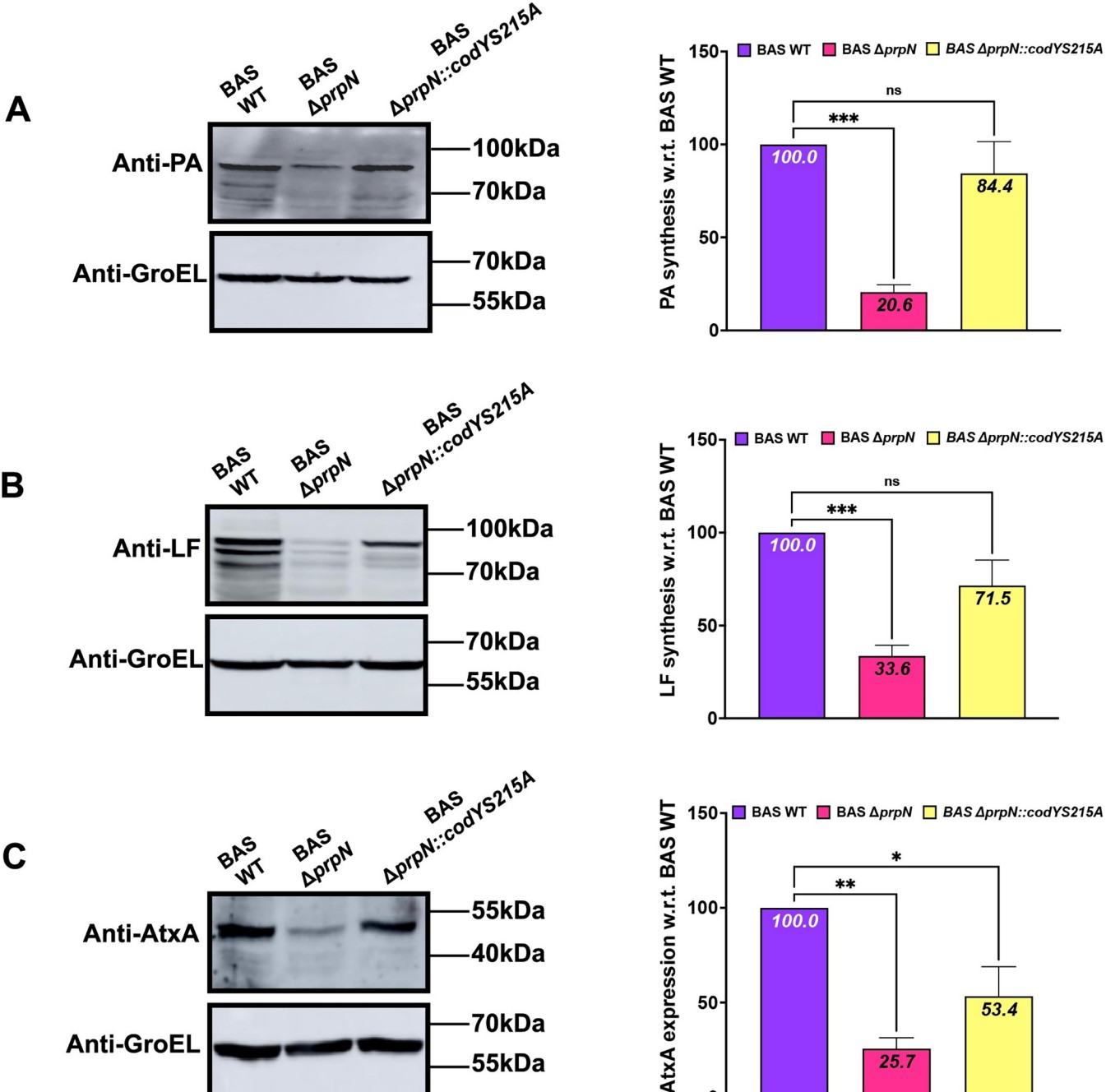

**Fig 8. Unphosphorylated CodY is an activator of toxin synthesis.** Representative immunoblots and bar graphs showing expression of PA (A), LF (B) and AtxA (C) in BAS WT, BAS Δ*prpN*, and BAS Δ*prpN* expressing CodY S215A (BAS Δ*prpN*::*codYS215A*). Immunoblots were analyzed as in Fig 5. Indicated MWs were derived from adjacent lanes containing PageRuler Prestained Protein Ladder, Thermo Scientific (Cat. No. 26616). Mean and standard deviation from minimum three independent experiments are shown in the bar graph. Asterisks indicate statistical significance of the data set calculated using one-way ANOVA followed by a *post hoc* test (Tukey test). *P* values < 0.05 were considered as statistically significant.

 d. Sporulation: inhibition of sporulation process,

 e. Toxin synthesis: ~80% decrease in toxin synthesis.

Apart from these, deletion of *prpN* showed defects in biofilm formation, while this defect was complemented in the BAS Δ*prpN*::*prpN* strain (S8 Fig). These were very interesting findings, but how is one ser/thr protein phosphatase regulating all the above-mentioned cellular pathways? To investigate and answer this important question, we checked the expression profile of PrpN protein and observed its expression level peaking at the onset of stationary growth phase (Fig 2E and 2F). In the model organism *B. subtilis*, the precise temporal expression of different STKs/STPs is linked to the regulation of specific cellular and developmental pathways such as biofilm formation, sporulation, and germination [59, 62, 93]. The reported data and our results suggested possible role of PrpN in transition and stationary phase genes regulation. To gain further insight, we focused the next part of our study on toxin synthesis as it is one of the most studied transition phase phenomena crucial for bacterial survival and virulence. While in the recent years considerable progress has been made in the host cellular pathways that are triggered by anthrax toxins and its mechanism of action resulting in the disease progression inside host, far less is known about the toxin synthesis paradigm in the life cycle of *B. anthracis*. In the present study, we have tried to explore this aspect and presented a novel ser/thr protein phosphatase mediated regulation of anthrax toxin expression via a global transcription regulator.

Considering the widespread ecological niches in which Bacillus is found, bacterial adaptation in response to environmental stimuli becomes imperative for its survival and growth. Modulation of bacterial transcriptome by global transcriptional regulators is one of the major ways by which it adapts to environmental fluctuations. Bacterial transcriptional regulators are often regulated by protein phosphorylation resulting in differential binding to the promoter regions and thereby expression/repression of various target genes [94–97]. CodY, a GTP binding global transcriptional regulator of over 500 genes including various stationary and virulence genes in *B. anthracis* was selected for our study [29]. Previous reports also indicated it as a potential target of ser/thr phosphorylation machinery and identified phosphorylation site at a serine residue (Ser215) in *B. subtilis* and *B. anthracis* [74, 98]. Since the expression of CodY is constant throughout the bacterial growth phase (S6 Fig), additional regulatory mechanisms such as protein phosphorylation might contribute to the modulation of CodY cellular activity [72]. This hypothesis was indeed reflected in our study as hyperphosphorylated CodY protein level (3 times) was detected in the *prpN* null mutant strain (Fig 6B and 6C). *In-vitro* ser/thr phosphatase assays of CodY purified from BAS Δ*prpN* strain incubated with PrpN further validated CodY as a target of PrpN (Fig 6D). Since protein phosphorylation at ser/thr residues is a reversible modification requiring specific ser/thr kinases and ser/thr phosphatases, we examined phosphorylation of CodY by the most well studied ser/thr kinase, PrkC in *B. anthracis* and our results confirmed it as a target of PrkC (Figs 6E, 6F and S5A). Though, the present study is limited to regulation of CodY phosphorylation by PrkC and PrpN, other ser/thr kinases (PrkD and PrkG) and ser/thr phosphatase (PrpC) might also contribute to this phenomenon. This assumption, however requires experimental evidence and further validation. Moving forward, our results also confirmed *in-vivo* phosphorylation of CodY specifically at serine 215 residue (Fig 7A and 7B). DNA binding assays of CodY phosphorylation site mutants revealed that CodY phosphorylation impedes its DNA binding activity with the *atxA* promoter gene (Figs 7C and S7C). A similar aberration in DNA binding activity is previously reported for the transition state transcriptional regulator- AbrB by phosphorylation at a serine residue in *B. subtilis* [99]. In *B. anthracis*, absence of *codY* resulted in severe AtxA and toxin protein synthesis defect thus indicating it as an activator of toxin synthesis, while AbrB is reported as a repressor of toxin synthesis [20, 28–30]. This indicates that a multifactorial regulatory network drives the expression of anthrax toxins and probably that is why toxin synthesis was only partially regained on overexpression of CodYS215A in the BAS Δ*prpN* strain (Fig 8).

# Toxin synthesis activation in *B. anthracis*

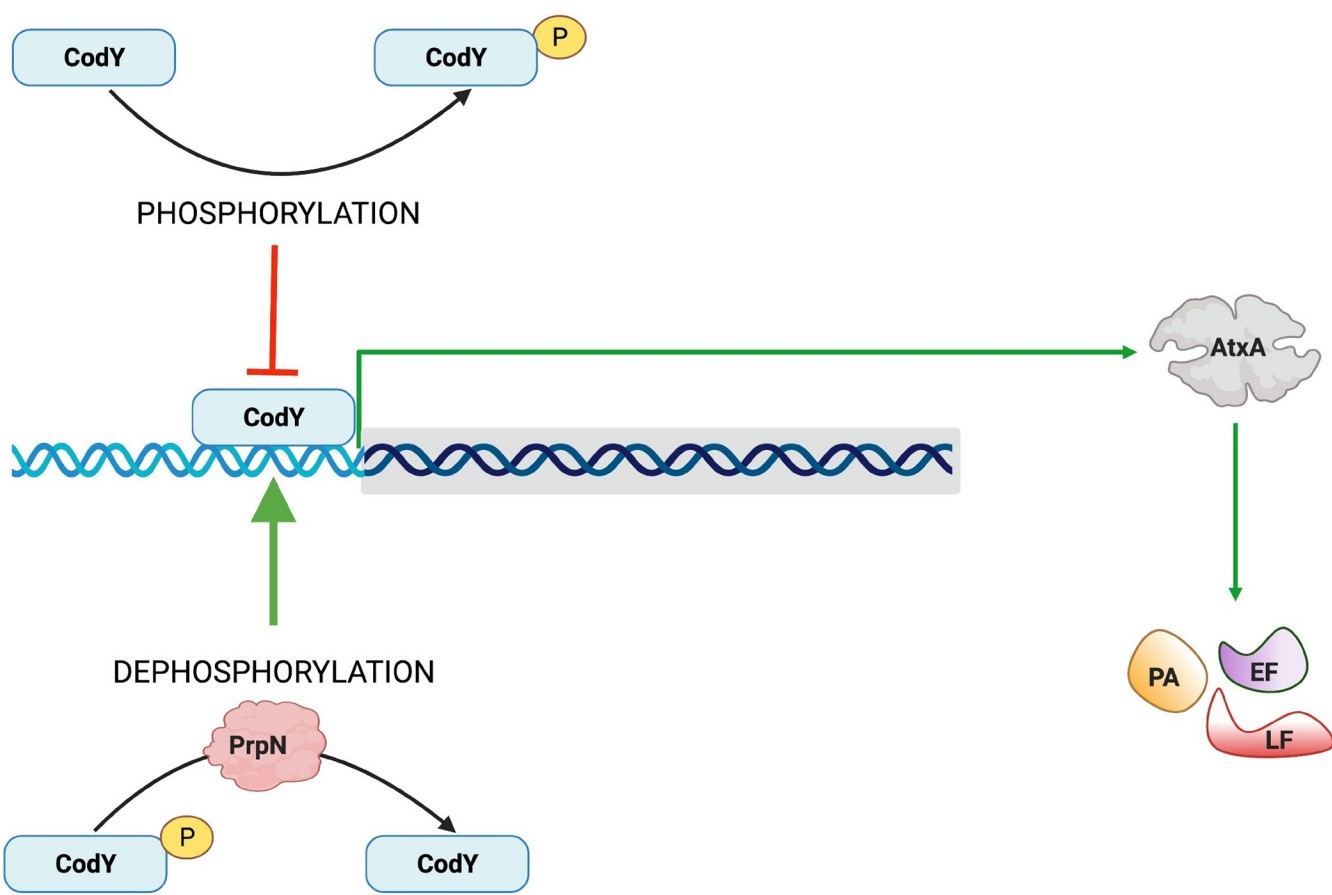

**Fig 9. Graphical Abstract. Schematic illustration of PrpN mediated toxin synthesis regulation via CodY.** Binding of unphosphorylated CodY protein to *atxA* promoter region activates AtxA expression, which thereby promotes toxin proteins (PA, LF and EF) synthesis. PrpN positively regulates anthrax toxin synthesis by dephosphorylation of CodY protein (Green arrow), while CodY phosphorylation abrogates its DNA binding ability to *atxA* promoter region (Red arrow) ultimately leading to downregulation of AtxA and anthrax toxin synthesis. The schematic was prepared by AG using BioRender.

Based on these studies and our results we conclude that not just CodY presence but the *in-vivo* CodY phosphorylation stoichiometry mediated by PrpN is also crucial for the activation of anthrax toxins (**Fig 9 Graphical Abstract**). Further, a recent report in *B. cereus* showed CodY mediated regulation of *clhAB2* operon involved in cell shape and bacterial chaining [100]. To study the functional implication of *in-vivo* CodY phosphorylation status in these processes, we examined the morphology of BAS Δ*prpN*::*codYS215A* strain and compared it with BAS WT and BAS Δ*prpN* strains. Morphological aberrations such as twisting and bending in the bacterial chains were considerably reduced in the BAS Δ*prpN*::*codYS215A* strain at both exponential and stationary growth phase (S9 Fig). This indicates that the regulation of CodY phosphorylation status by PrpN is critical for rod-shape maintenance of metabolically active vegetative bacterial cells in *B. anthracis*. Moreover, though the mechanistic aspect and PrpN targets in context of sporulation were not explored in the present study, a previous study in *B. anthracis*

demonstrated CodY as a repressor of sporulation [72]. This suggests that the effect of PrpN on sporulation depends on other regulators or factors downstream of CodY in the sporulation cascade.

Further, since CodY HTH motif is highly conserved among CodY homologs and a direct role of CodY in virulence and toxin synthesis is widely reported in various Gram-positive pathogens, the present study can be extended to other bacterial strains as well and might suggest a similar phosphorylation dependent modulation of CodY DNA binding ability and thereby its target gene regulation [75, 101, 102]. Most importantly, as lower toxin gene expression accounts for lower virulence, the strain used in this study might also be attenuated for virulence. Thus, it can be further explored for the generation of novel vaccine strain against anthrax and help in overcoming the shortcomings of the currently available vaccines by providing the required balanced level of virulence attenuation along with retention of protective ability.

## Material and methods

### Ethics statement

All the animal experiments were conducted in accordance with the guidelines mentioned by the Institutional Animal Ethics Committee (IAEC) of Department of Zoology, University of Delhi, India. Balb/c mice were used for raising polyclonal antibodies used in the manuscript and animal handling was done in accordance with the IAEC approved letter with the protocol number- DU/ZOOL/IAECR/2019.

### Bacterial strains and growth conditions

*Escherichia coli* (*E. coli*) strains DH5α was used for cloning and BL21(DE3) was used for the recombinant proteins production. For electroporation in bacillus cells, the plasmids were further passed through *E. coli* SCS110 (*dam⁻/dcm⁻*). Luria Bertani (LB) broth (Difco) and LB Agar (Difco) was used for growing *E. coli* strains. *B. anthracis* Sterne (BAS) strains were either grown in LB broth/Agar (Difco) or Nutrient Broth Yeast (NBY) Extract medium (0.8% nutrient broth, 0.3% yeast, and 0.5% glucose based on the experimental requirement. Ampicillin and kanamycin antibiotics were used for selections in *E. coli* and *B. anthracis* at a final concentration of 100 μg/mL and 25 μg/mL, respectively. Cultures were grown at 37˚C with proper aeration (1:5 head space) and shaking at 200 rpm. For toxin synthesis, strains were grown in NBY medium containing 1% sodium bicarbonate (NaHCO$_3$) for 4 hours at 37˚C. To examine the growth kinetics, glycerol stocks of bacillus strains were streaked on LB agar plate and an isolated colony was used for primary inoculum. Log phase bacillus cultures were used to initiate secondary cultures at a starting $A_{600nm}$ of 0.02. $A_{600nm}$ was monitored at 2-hour interval until 14 hours. For stress experiments, either 1 M NaCl (salt stress) or 2.5 mM H$_2$O$_2$ (oxidative stress) was introduced when the cultures attained an $A_{600nm}$ of approximately 0.1 (lag to log phase transition). Description of plasmids and strains used in the study are provided in S2 and S3 Tables, respectively.

### Operon prediction

Genomic arrangement of *prpN* gene (BAS0539) was fetched from NCBI Database and two intergenic primer sets were synthesized. *rpoB* and *prpN* gene specific primers were also used to confirm successful cDNA preparation. PCR reactions were set in a reaction volume of 20 μL containing 2 μL template, 4 μL 5X HF buffer, 2 μL dNTP mix (2.5 mM stock), 2 μL primer mix (10 μM stock) and 0.2 U Phusion DNA Polymerase (Thermo Scientific). DNA Amplification

was performed in a thermocycler (Bio-Rad, T100) with the following parameters: initial denaturation at 95˚C for 10 mins, followed by 35 cycles of denaturation at 95˚C for 30 s, annealing at 54˚C for 1 min, extension at 72˚C for 1 min, and a final extension at 72˚C for 10 mins. The amplified products were resolved on an agarose gel and stained using EtBr. Primers details are listed in S1 Table.

### PrpN knockout and complemented strain generation

The single gene knockout strain of *prpN* in *Bacillus anthracis* Sterne (BAS Δ*prpN*) strain was generated using the method described earlier [47]. Briefly, two single crossover plasmids (pSC) derived from the temperature sensitive shuttle vector pHY304 were used in a sequential order for the insertion of loxP sites in the surrounding region of *prpN* gene (BAS0539). Another, temperature sensitive plasmid pCrePAS2, expressing Cre recombinase was then used for the subsequent excision of the DNA region between the two inserted loxP sites, hence deleting the *prpN* gene from the BAS genome. Positive colonies were confirmed using *prpN* gene flanking and internal primers. For complemented strain (BAS Δ*prpN*::*prpN*), *prpN* gene along with 400 bp upstream region was amplified. The amplified product containing KpnI and SmaI sites was digested and ligated in pYS5 shuttle vector [103]. Positive clones were confirmed by sequencing and unmethylated plasmid was electroporated in BAS WT strain and the transformants were selected on LB agar plates supplemented with kanamycin (25 μg/mL). For confirmation, colony PCR was done using pYS5 specific primers. Primers details are listed in S1 Table.

### Site-Directed Mutagenesis (SDM) for CodY mutants preparation

CodY Serine215 site was mutated to alanine residue (CodYS215A) and glutamate residue (CodYS215E) following the instructions provided by QuikChange II XL Site-Directed Mutagenesis Kit (Agilent). CodY cloned in pPROEXHTc vector was used as a template for the SDM reactions. The positive clones are named as pPROEXHTc-*codYS215A* and pPROEXHTc-*codYS215E* in the study. For pYS5-*codYS215A*, pYS5-*codY* plasmid served as a template and a hexa-histidine tag was introduced in the reverse primer for protein purification. All the clones were confirmed by sequencing and primer details are provided in S1 Table.

### Cloning, expression, and protein purification

The genes encoding *prpN* (BAS0539), *prpC* (BAS3714), *atxA* (GBAA_pXO1_0146), and *codY* (BAS3679) were cloned in pPROEXHTc vector, which produces proteins having an N-terminal hexa-histidine (His$_6$) tag. *E. coli* BL21 (DE3) cells were transformed with these recombinant plasmids for protein production and the recombinant His$_6$-tagged proteins were purified using the method described earlier [41]. For purification of PrkCcat-GST tagged protein, the gene encoding the catalytic domain of *prkC* was cloned in pGEX5x3 vector and the recombinant protein was purified using the method described earlier [104]. For purification of CodY protein from bacillus strains, the genes encoding *codY* and *codYS215A* were cloned in pYS5 vector. A His$_6$ tag was incorporated in the reverse primer for the production of CodY-His$_6$ tag protein. Bacillus strains harboring this plasmid (pYS5-*codY*$_{His6}$ or pYS5-*codYS215A*$_{His6}$) were grown at 37˚C at 200rpm. The cultures were harvested and the cell pellets were washed thrice with 1X PBS followed by resuspension in lysis buffer [50 mM Tris-HCl (pH 8.0), 300 mM NaCl, 1 mM PMSF, 10% glycerol, 1X protease inhibitor cocktail (Roche Applied Science) and 1X phosphatase inhibitor cocktail (Sigma-Aldrich)]. The cells were sonicated (25% amplitude, 30:20 ON:OFF, total time 8 mins) and the cell lysates were centrifuged at 20,000 × g at 4˚C for 30 mins. The supernatant containing recombinant proteins was incubated with Ni-NTA

agarose resin (Qiagen) preincubated with Buffer A [50 mM Tris-HCl (pH 8.0) and 300 mM NaCl] for 3 hours. The resins were washed with Buffer B [50 mM Tris-HCl (pH 8.0), 300 mM NaCl, 1 mM PMSF and 10 mM imidazole] and Buffer C [50 mM Tris-HCl (pH 8.0), 1 M NaCl, 1 mM PMSF and 20 mM imidazole] for 45 mins each. Finally, the protein was eluted in Buffer D [50 mM Tris-HCl (pH 8.0), 150 mM NaCl, 1mM PMSF and 300 mM imidazole]. The purified proteins were further dialyzed and estimated using BCA (Thermo-Fisher Scientific) for downstream assays. Primers and plasmid constructs details are listed in S1 and S2 Tables, respectively.

## Circular dichroism (CD) spectroscopy

CD spectroscopy was performed to detect the structural integrity and mutation induced folding properties of CodY proteins used in the present study (CodY native, CodYS215E and CodYS215A). The CD spectrum of purified proteins was recorded on JASCO, J-815 CD spectrophotometer and the data was plotted in a graph to examine structural properties of these proteins.

## Polyclonal antibody generation

30 μg of PrpN, AtxA and CodY proteins were emulsified in Freund's complete adjuvant (Sigma-Aldrich) and injected subcutaneously in three BALB/c mice. Two boosters of 15 μg each (emulsified in Freund's incomplete adjuvant) were given post 15 days for the stimulation of antibody production. All mice were bled 15 days post the 2nd booster injection and serum was collected. Antibody titer was determined by indirect ELISA.

## Bacillus whole cell lysates preparation and immunoblot analysis

Harvested bacterial cells were resuspended in a lysis buffer containing 50 mM Tris-HCl (pH 8.0), 1 mM EDTA, 100 mM NaCl, 1 mM PMSF, 10% glycerol, and 1X protease inhibitor cocktail (Roche Applied Science). The cells were sonicated at 25% amplitude, 30:20 ON:OFF, total time 8 mins and protein was estimated using BCA (Thermo Fisher Scientific). For immunoblots, equal amounts of whole cell lysates (20–40 μg) were resolved on 10% or 12% SDS-PAGE along with a prestained protein ladder and transferred to a NC membrane (Millipore). Following this, the membrane was blocked for 1 hour in 5% BSA prepared in PBST (phosphate buffer saline with 0.1% Tween 20). Membrane was washed thrice with PBST and incubated with a primary antibody [anti-PrpN (1:20,000) or anti-CodY (1:50,000) or anti-AtxA (1:50,000)] for 1 hour. To prevent non-specific antibody binding 1% BSA was included in the primary antibody dilution. The blots were washed 5 times with PBST and incubated with anti-mouse IgG antibody conjugated with horseradish peroxidase (1:20,000 –Cell Signaling Technology, Cat. No. 7076S) for another 1 hour. The blot was again washed 5 times with PBST and finally developed using SuperSignal West Pico PLUS/Femto PLUS Chemiluminescent substrate (Thermo Fisher Scientific) and quantified with the luminescent image analyzer (Amersham Imager600 or ImageLab6.0.1). The same blot was stripped and probed similarly using anti-GroEL (1:200,000), used as a loading control. Detection of phosphorylated serine residues was done using commercial anti-phosphoserine antibody (1:2000, Abcam, Cat. No. ab9332) and anti-rabbit IgG conjugated with horseradish peroxidase (1:20,000 –Cell Signaling Technology, Cat. No. 7074S) was used for secondary antibody. For phosphoprotein immunoblots, PBST was replaced by tris buffer saline (TBS) containing 0.05% Tween 20 (TBST). The data were plotted using GraphPad Prism.

## Phos-tag western blotting to examine *in-vivo* stoichiometry of CodY protein phosphorylation at serine215 residue

Native CodY and CodYS215A with a C-terminus hexa-histidine tag were produced in BAS WT and BAS Δ*prpN* strains using pYS5 shuttle vector. 2 μg of purified proteins was resolved on SuperSep Phos-tag (50 μmol/L), 12.5% gel (Cat. No. 195–17991) for separation of phosphorylated and unphosphorylated CodY using the method described earlier [105]. These proteins were also resolved on 12.5% SDS-PAGE to examine the mobility pattern of CodY in the absence of phos-tag. The gels were transferred onto a NC membrane and probed using anti-CodY antibody following the protocol mentioned above.

## *In- vitro* kinase assay

In-vitro kinase assay was performed using the method described earlier with slight modifications [38]. Briefly, 100 ng GST tagged PrkC catalytic domain (PrkCcat-GST) was first preactivated with 1 mM ATP. 2 μg CodY-His$_6$ was then incubated with autophosphorylated PrkCcat-GST in kinase buffer [20 mM HEPES (pH 7.2), 1 mM DTT, 10 mM MgCl$_2$ and 10 mM MnCl$_2$) containing 1 mM ATP at 25°C for 30 minutes. Reactions were terminated by 1X SDS sample buffer followed by boiling at 95°C for 10 minutes. CodY alone and CodY with ATP were used as controls and the protein samples were resolved on 12.5% SDS-PAGE. The gel was transferred onto a NC membrane and probed using anti-phosphoserine antibody to analyze the phosphorylation status of CodY. Additionally, PrkC mediated phosphorylation of CodY was also examined by staining the gel with Multiplexed Proteomics Phosphoprotein Gel Stain Kit #2 containing Pro-Q Diamond and SYPRO Ruby gel stains (Invitrogen, Thermo-Fisher Scientific). Images were captured using Bio-Rad Imager.

## Analysis of anthrax toxin proteins—PA and LF

Bacillus strains were grown in NBY broth and the cells were harvested at 12,000g, resuspended in lysis buffer and sonicated (as mentioned in above section). The whole cell lysates were estimated by BCA (Thermo-Fisher Scientific) and used for examining toxin proteins synthesis, while the supernatant was filtered through 0.22 μm filter assembly and the protein was precipitated using 20% trichloroacetic acid (TCA). The precipitated protein was resuspended in a resuspension buffer [1% SDS, 10% glycerol, 10 mM Tris-HCl (pH 6.8), 2 mM EDTA, 100 mM DTT and 1X protease inhibitor cocktail (Roche Applied Science)] and used further for assessing the secretion of PA and LF proteins. Anti-PA (1:50,000) and Anti-LF (1:50,000) raised in rabbit were used for immunoblot analyses.

## Spore preparation and quantification

Bacillus spores were prepared using the method described earlier [106]. Half of the sample was heated at 65°C to kill any residual vegetative cells. Serial dilutions of untreated and heat-treated samples were plated on LB agar plates. The plates were incubated overnight at 37°C and colonies were counted (CFU) for estimation of sporulation efficiency and total spore count.

$$\text{Total spore count } (\%) = [\text{CFU per mL (heat treated)} \div \text{CFU per mL (untreated)}] \times 100$$

$$\text{Sporulation efficiency } (\%) = [\% \text{ Test total spore count (BAS } \Delta prpN \text{ or BAS } \Delta prpN :: prpN \div \% \text{ Control total spore count (BAS WT)}] \times 100$$

## Phosphatase assay

Phosphatase activity of purified PrpN protein was determined spectrophotometrically using para-nitrophenyl phosphate-pNPP (Sigma-Aldrich) as substrate in a 96-well plate. The hydrolysis of pNPP to para-nitro phenolate was measured at 405 nm. The experiment was performed using different pH buffers and time intervals for optimizing the assay conditions. Also, since PPM phosphatases requires metal ions (magnesium or manganese) for phosphatase activity, the experiment was carried out in the presence of different metal ions [5 mM magnesium ($Mn^{+2}$), 5 mM manganese ($Mg^{+2}$), 5 mM zinc ($Zn^{+}$) and 5 mM calcium ($Ca^{+2}$)]. *B. anthracis*, ser/thr protein phosphatase–PrpC was taken as a positive control and various negative controls were taken (only buffer, only protein, only substrate and EDTA/EGTA). Once all the required standardization was done, an equal amount of PrpN and PrpC protein (1 μg) was added in the wells containing assay buffer [50 mM Tris-HCl (pH 8.0), 5 mM $MnCl_2$, 0.01% β-mercaptoethanol] and 5 mM pNPP. Absorbance was recorded at 405nm after 30 minutes using Epoch multi-plate reader (Biotek) and the calculated PrpN phosphatase activity was plotted using GraphPad Prism. Following this, PrpN ser/thr specific phosphatase activity was measured following the protocol provided by Serine/Threonine Phosphatase Assay System kit (Promega). Briefly, PrpN protein was added in a reaction volume containing assay buffer and synthetic ser/thr phosphopeptides as substrate. The free phosphate in the reaction was measured at 600nm using Epoch multi-plate reader (Biotek). PrpN phosphatase activity was calculated using a standard phosphate curve and the values were plotted in a Michaelis-Menten curve to determine the enzyme kinetics parameters. Additionally, the synthetic ser/thr phosphopeptide used in the above assay was replaced with CodY protein purified from BAS Δ*prpN* strain and the generated phosphate in the reaction was estimated.

## Phase contrast microscopy

Bacterial cells at different growth phase were harvested and processed using the method described earlier [41]. The cells were visualized under Zeiss Axio Imager Z2 upright microscope (100x/1.4 oil DIC objective) and images were procured using Axiocam 506 color camera attached to the microscope. All the images were processed using ZEN 2 Pro software. Sporulation kinetics of BAS WT and BAS Δ*prpN* strains were studied by live-cell phase-contrast microscopy using agarose pads supplemented with sporulation medium [8g of LB broth/L supplemented with 85.5 mM NaCl, 0.025 mM $ZnSO_4$, 0.6 mM $CaCl_2$, 0.3 mM $MnSO_4$, 0.8 mM $MgSO_4$, and 0.02 mM $CuSO_4$] and pH was set to 6.0. Agarose pads were prepared using the AB gene frame on frosted glass slides (Corning Micro slide Frosted; 75*25 mm) [41, 92]. Low melting agarose (3%) supplemented with sporulation medium was evenly poured on the glass slides and left for solidification. Exponentially growing cultures of both the strains were diluted to an $A_{600nm}$ = 0.035 and 2 μL of this suspension was spread evenly on the agarose pads. These were then incubated at 30˚C and images were captured at 24, 48, 72, 96 and 144 hours using an inverted microscope.

## Electron microscopy (Scanning and transmission)

Exponentially growing vegetative cells of BAS WT, BAS Δ*prpN* and BAS Δ*prpN*::*prpN* strains were processed for scanning electron microscopy using the protocol described earlier [41]. Cells were visualized under Zeiss Scanning Electron Microscope EVO LS15 at 20 KV and analyzed using Smart SEM software.

Ultrastructural details of BAS WT, BAS Δ*prpN* and BAS Δ*prpN*::*prpN* spores were examined using transmission electron microscopy. Samples were processed by two different methods to visualize vegetative sporulating cells population using the protocols described earlier

[41, 107]. Briefly, for pure spore suspension, sporulation medium was removed after sporulation completion and the harvested spores were kept in water for three days to ensure vegetative cell lysis. These were again washed thrice with water followed by resuspension in water to remove vegetative cell debris and fixed for TEM analysis. To visualize intact cells in the sporulation medium, the bacterial cells were harvested and washed thrice with 0.85% saline, resuspended in the same and fixed. Sectioning was done using Leica UC6 ultra-cut and the sections were then observed in FEI Tecnai G2 Spirit at 200 KV.

### Confocal microscopy

Morphological characteristics and septal analysis of BAS WT and BAS Δ*prpN* strains were done by labeling the cells with the membrane stain FM4-64 (Thermo Fisher Scientific) and observing the live cells by fluorescence microscopy. 2 μL of BAS WT and BAS Δ*prpN* exponentially growing culture diluted to $A_{600nm}$ = 0.035 were evenly spread on LB agarose pads supplemented with 1μg/mL FM4-64 dye for staining the cell membrane and incubated at 37°C overnight. Image acquisition was done using Leica TCS SP8 confocal laser scanning microscope (63X oil immersion objective).

### RNA extraction and quantitative real time PCR

BAS WT and BAS Δ*prpN* strains were grown in triplicates up to the early stationary phase and RNA was extracted by hot lysis method described previously [41]. The purified RNA samples were used to prepare the cDNA using a first-strand cDNA synthesis kit (Thermo-Fisher Scientific). To analyze the expression of *atxA* in BAS WT and BAS Δ*prpN* strains, the corresponding cDNA was used along with *atxA* gene-specific primers and SYBR Green master mix (Roche Life Science), following the instructions provided in the manual. Reactions were prepared in triplicates along and run in a LightCycler 480 Instrument II (Roche Life Science). A no template control was also included to check gDNA contamination and data normalization was done with respect to *rpoB* ((DNA-directed RNA polymerase β subunit) expression level [41]. Primer details are listed in S1 Table.

### Electrophoretic mobility shift assay

4 nM *atxA* promoter gene was used as probe and incubated with increasing amounts of CodY proteins (1 μM-12 μM) in a 20 μL reaction volume containing 1X binding buffer [20 mM Tris-HCl (pH 7.2), 50 mM KCl, 1.5 mM MgCl₂, 5% glycerol, 0.5 mM EDTA, 1 mM DTT, 0.05% Nonidet P40]. The reaction was incubated at 30°C for an hour and resolved by electrophoresis using a 6% Native-PAGE. The same protocol was followed in the presence of 1 mM GTP and 1 mM BCAA (Isoleucine, Leucine and Valine). A fluorescence-based EMSA Kit was used for sequential detection of DNA by SYBR Green and protein by SYPRO Ruby in the same gel using the instructions provided in the manual (Invitrogen, Thermo-Fisher Scientific). Images were captured using Bio-Rad Imager.

### Biofilm formation

Biofilm formation and efficiency was measured using the method described earlier with some modifications [108]. Briefly, BAS WT, BAS Δ*prpN* and BAS Δ*prpN* strains were grown overnight in LB medium and secondary cultures for each strain (1.5% vol/vol) were initiated in 6-well plates containing 5mL LB medium. The plates were incubated at 37°C for 72 hours without shaking for biofilm formation and quantified by crystal violet assay as described previously [109].

## Statistical analysis

Unless mentioned otherwise, a minimum of three independent experiments (biological) repeated thrice (technical) were done to ensure data reproducibility. GraphPad Software (Prism 6) was used to plot the data and significance of the results was analyzed using two-tailed Student's *t* test or one-way ANOVA followed by a *post hoc* test (Tukey test). *P* values < 0.05 were considered as statistically significant.

## Densitometer analysis

Unless mentioned otherwise, all immunoblots densitometer analysis were done using Amersham Imager600.

## Fig preparation

All illustrations/schematics were prepared using BioRender and figures were prepared using ImageJ and Adobe Illustrator.

## Supporting information

**S1 Fig. Multiple sequence alignment and structural analysis of PrpN.** (A) Multiple sequence alignment of PrpN in the most studied Bacillus genus strains: *B. cereus*, *B. thuringiensis* and *B. subtilis* using Clustal Omega. "*" denotes perfect alignment, ":" denotes strong similarilty and "." denotes weak similarity. (B) Structural overview of PrpN protein predicted using I-TAS-SER online server (Iterative threading assembly refinement). Primary amino acid sequence and structure of PrpN protein showing helix (pink colour), β-strands (yellow colour) and coils (blue colour). (C) PrpN structure depicting the phosphatase domain [metal-dependent protein phosphatase (MPP) family member] in cyan colour. Highly conserved metal-binding sites are indicated in the right panel.
(TIF)

**S2 Fig. Phylogenetic analysis of PrpN in firmicutes.** A phylogenetic tree representing the evolutionary relationship of PrpN in above-mentioned organisms. It was generated using Neighbor-Joining analysis conducted in MEGA XI. The tree is drawn to scale with branch lengths. This analysis involved 45 amino acid sequences.
(TIF)

**S3 Fig. Knockout and complemented strain confirmation.** (A) Schematic representation of BAS Δ*prpN* and BAS Δ*prpN*::*prpN* strain generation. (B) Agarose gel showing PCR products amplified by *prpN* gene specific primers using indicated strains gDNA as template for strains' confirmation at gene level. Ladder- 100 bp DNA Ladder H3 RTU (GeneDirex, Cat. No. SD003-R600). (C) Whole cell lysates of indicated strains were loaded in equal amount and probed using anti-PrpN and anti-GroEL for strains' confirmation at protein level. Purified recombinant PrpN proteins with hexa-histidine tag was used as a positive control. Ladder-PageRuler Prestained Protein Ladder, Thermo-Scientific (Cat. No. 26616).
(TIF)

**S4 Fig. Quantification of bacterial cell length.** Bacterial cell length was measured using ImageJ software and the values were plotted as box and violin graph using GraphPad Prism software. All the data points (N = 300) are indicated in the graph and the corresponding mean cellular length and coefficient of variation is depicted in a table.
(TIF)

**S5 Fig. CodY is a target of PrkC.** (A) *In-vitro* kinase assay was performed by incubating 2 μg CodY with 100 ng autophosphorylated PrkCcat. CodY alone and CodY incubated with ATP were taken as controls. The samples were resolved on 12% SDS PAGE and phosphorylation was visualized by using Pro-Q diamond phospho specific gel stain (upper panel) and SYPRO Ruby stain was used to visualize resolved proteins. (B) *In-vivo* PrkC mediated phosphorylation of CodY. CodY purified from indicated strains was loaded in equal amount and probed using anti-phosphoserine and anti-CodY. Indicated MWs were derived from adjacent lanes containing PageRuler Prestained Protein Ladder, Thermo Scientific (Cat. No. 26616).
(TIF)

**S6 Fig. Expression of CodY at different growth phases.** (Left panel) Representative immunoblot showing growth dependent differential expression of CodY in BAS WT strain. Equal amount of protein lysates prepared from different growth phases–(i) lag phase, (ii) exponential phase, (iii) early stationary phase and (iv) late stationary/sporulation initiation phase was loaded and probed using anti-PrpN and anti-GroEL. Lane (v) indicates purified recombinant CodY protein. Ladder- PageRuler Prestained Protein Ladder, Thermo-Scientific (Cat. No. 26616). (Right panel) Densitometer analysis were done using Amersham Imager600 software and the corresponding CodY/GroEL ratio were plotted in a bar graph using GraphPad Prism. Densitometer readings calculated from three experiments executed independently are shown in the bar graph.
(TIF)

**S7 Fig. Structural analysis of CodY mutants and EMSA in the presence of effectors.** (A) Schematic illustration of the strategy followed for CodY mutant generation. (B) Coomassie stained SDS-PAGE of recombinant native CodY, CodYS215A and CodYS215E proteins (left panel). Superimposed CD spectrum of native and mutant CodY proteins (right panel). (C) Electrophoretic Mobility-Shift Assay using SYBR Green and SYPRO Ruby stains in the presence of effector molecules. Increasing amounts (1, 1.5, 3, 4, 5, 6, 8, 10 and 12μM) of native CodY, CodY S215E and CodY S215A proteins were added in a binding reaction containing 4 nM *atxA* promoter region as probe in the presence of GTP and BCAA. Lane 1 represents only DNA control. Upper gel image represents DNA bands stained using SYBR Green and lower image represents protein bands stained using SYPRO Ruby.
(TIF)

**S8 Fig. Effect of *prpN* deletion on biofilm formation.** (A) Representative images of BAS WT, BAS Δ*prpN* and BAS Δ*prpN*::*prpN* biofilm formation in a 6-well plate. (B) Biofilm formation efficiency of these strains was estimated using crystal violet stain and were plotted w.r.t. BAS WT. Mean and standard error mean from six independent experiments are shown in the bar graphs. Statistical Analysis: Asterisks indicate statistical significance of the data set calculated using two-tailed Student's *t* test. * corresponds to $p<0.05$; ** corresponds to $p<0.01$; *** corresponds to $p<0.001$ and **** corresponds to $p<0.0001$.
(TIF)

**S9 Fig. Unphosphorylated CodY is crucial for vegetative rod-shape morphology.** Representative phase contrast microscopy images of BAS WT, BAS Δ*prpN* and BAS Δ*prpN*::*codYS215A* strains at different time points. Scale bars are depicted in the images.
(TIF)

**S1 Table. List of primers used in this study.**
(PDF)

**S2 Table. List of plasmids used in this study.**
(PDF)

**S3 Table. Bacterial strains used in this study.**
(PDF)

## Acknowledgments

We thank electron microscopy staff at AIIMS, Delhi for their help in SEM and TEM imaging and confocal microscopy staff at CSIR-IGIB, Mathura Road, Delhi for their help in confocal imaging. We would also like to thank Mr. Hem Narayan Sharma, animal house staff, Department of Zoology, University of Delhi for help in antibody preparation.

## Author Contributions

**Conceptualization:** Aakriti Gangwal, Neha Dhasmana.

**Data curation:** Aakriti Gangwal, Nitika Sangwan, Neha Dhasmana, Nishant Kumar, Lalit K. Singh, Stephen H. Leppla, Yogendra Singh.

**Formal analysis:** Aakriti Gangwal, Nitika Sangwan, Neha Dhasmana, Yogendra Singh.

**Funding acquisition:** Stephen H. Leppla, Yogendra Singh.

**Investigation:** Aakriti Gangwal, Nitika Sangwan, Neha Dhasmana, Stephen H. Leppla, Yogendra Singh.

**Methodology:** Aakriti Gangwal, Nitika Sangwan, Neha Dhasmana, Nishant Kumar, Chetkar Chandra Keshavam, Lalit K. Singh, Ankur Bothra, Ajay K. Goel, Andrei P. Pomerantsev.

**Project administration:** Aakriti Gangwal, Stephen H. Leppla, Yogendra Singh.

**Resources:** Stephen H. Leppla, Yogendra Singh.

**Software:** Aakriti Gangwal, Nishant Kumar, Chetkar Chandra Keshavam.

**Supervision:** Ajay K. Goel, Stephen H. Leppla, Yogendra Singh.

**Validation:** Aakriti Gangwal, Ankur Bothra.

**Visualization:** Aakriti Gangwal.

**Writing – original draft:** Aakriti Gangwal.

**Writing – review & editing:** Aakriti Gangwal, Nitika Sangwan, Neha Dhasmana, Nishant Kumar, Chetkar Chandra Keshavam, Lalit K. Singh, Ankur Bothra, Ajay K. Goel, Andrei P. Pomerantsev, Stephen H. Leppla, Yogendra Singh.

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
