## [Decision Letter · Decision Letter 0]

27 Dec 2021

Dear Mr. Singh,

Thank you very much for submitting your manuscript "Role of serine/threonine protein phosphatase PrpN in the life cycle of Bacillus anthracis" for consideration at PLOS Pathogens. As with all papers reviewed by the journal, your manuscript was reviewed by members of the editorial board and by several independent reviewers. In light of the reviews (below this email), we would like to invite the resubmission of a significantly-revised version that takes into account the reviewers' comments.

Please respond to all reviewer comments. You must include experiments that address (1) the potential impact of PrkC on CodY phosphorylation, (2) whether CodY is a direct PrpN target, (3) CodY binding to DNA in the presence of GTP and branched chain amino acids, and (4) the effect of the CodY S215A mutation on toxin gene expression. 

We cannot make any decision about publication until we have seen the revised manuscript and your response to the reviewers' comments. Your revised manuscript is also likely to be sent to reviewers for further evaluation.

Sincerely,

Theresa M. Koehler

Associate Editor

PLOS Pathogens

Christoph Tang

Section Editor

PLOS Pathogens

Kasturi Haldar

Editor-in-Chief

PLOS Pathogens

orcid.org/0000-0001-5065-158X

Michael Malim

Editor-in-Chief

PLOS Pathogens

orcid.org/0000-0002-7699-2064

Reviewer's Responses to Questions

**Part I - Summary**

Reviewer #1: This straightforward genetic study characterizes the deletion mutant of gene BAS_0539, a putative ser/thr phosphatase of Bacillus anthracis. Whereas characterization of predicted Ser/Thr kinases (three genes in B. anthracis, PrkC, PrkD, PrkG) and phosphatase PrpC have been evaluated previously, the current article is the first to evaluate a second phosphatase, designated PrpN. Signal transduction via phosphorylation on protein Ser/Thr residues is still relatively poorly understood in bacteria despite several studies indicating the importance of such regulatory systems. This study shows that PrpN substantial influences a variety of phenotypes central to B. anthracis physiology and virulence, including effects on cell division and growth, sporulation, and toxin production. Importantly, phosphorylation of CodY appears to serve as a primary mediator of these cellular functions, as shown through demonstration of phospho-mimicking and -ablative mutants. Overall, the paper flows logically, is easy to follow and understand, experiments are completed with rigor, and its conclusions are substantiated by the results. I have only a few minor suggestions to provide.

Reviewer #2: The paper by Ganwal et al describes very interesting results on the characterization of a new phosphatase that controls virulence factor production and sporulation in Bacillus anthracis. The authors have identified the key regulator CodY as a possible target of PrpN since its level of phosphorylation is controlled by PrpN. The proposed PrpN-dependent dephosphorylation of CodY controls the synthesis of the key regulator of virulence AtxA and then of toxin production. However, these interesting results should be completed (see below). The paper should also be improved and the results should be more carefully discussed.

Reviewer #3: Here the authors demonstrate a role for PrpN in regulating multiple cellular functions in B. anthracis. They show that loss of PrpN causes dramatic changes to cell morphology and spore formation. They also report a decrease in toxin production of both PA and LF. Based on homology to B. subtilis they test if PrpN is required for dephosphorylation of CodY in B. anthracis and show that PrpN likely dephosporylates CodY. They then show that mutation of the site of phosphorylation impacts the DNA binding capacity of CodY in vitro. They also find show that mutation of the CodY S215A decreases the presumed effect of phosphorylation of CodY on toxin production however this only explains part of the effect of loss of prpN on toxin expression.

**Part II – Major Issues: Key Experiments Required for Acceptance**

Reviewer #1: None.

Reviewer #2: What is the kinase involved in the phosphorylation of CodY ? This point is never discussed in the text. Since the authors have recently published a manuscript on PrkC of B. anthracis and have a prkC mutant, the impact of PrkC inactivation on the level of phosphorylation of CodY in vivo should be tested. A phosphorylation test of CodY by PrkC in vitro could be also done.

The authors showed that the level of phosphorylation of CodY increased in the prpN mutant in B. anthracis. However, they did not demonstrate that CodY is a direct PrpN target and that the site of phosphorylation of CodY in Bacillus anthracis is Ser215. They should complete their study by an analysis of the purified CodY they obtained from B. anthracis (WT and ∆prpN strains) by Mass spectrometry to confirm that this serine is the phosphosite targeted by PrpN. Alternatively, a CodYSer215A-His6 could be expressed in B. anthracis and purified by the same strategy than the wild-type copy of CodY. To confirm that the effect of PrpN on CodY is direct, the addition of the purified PrpN to the CodY purified in vivo followed by a western-blot with an antipSer should be done. In addition, to estimate the purification of CodY, a gel should be added to western blot in Figure 7B. Finally, we would also suggest to use PhosTag gel combined with the Anti-CodY antibody. This PhosTag system, which leads to a separation of phosphorylated and unphosphorylated forms of proteins will allow to estimate more precisely the proportion of the CodY protein phosphorylated in your different strains.

The impact of prpN inactivation on other phenotypes related to CodY should be tested (motility, biofilm formation, heme as iron source…). Since CodY controls an operon involved in autolysis and chain separation in Bacillus cereus (Huillet et al, Plos one 2016), we would suggest to analyze the morphology of the strains carrying a phosphoablative mutation of CodY you have obtained.

Reviewer #3: 1. The EMSA shift data showing mutants of CodY fails to bind DNA is convicing. However there are a few issues with it first the binding affinities are nearly 100 times higher than other reported binding affinities. This could be in part due to not using GTP and Branched chain amino acids or use of SYBR green vs. P32 labeled probes.

2. Second they do not test binding in the presence of GTP or branch chained amino acids which enhance DNA binding of CodY. It may be important to know how DNA binding is affected in the presence of the signals that promote DNA binding.

3. Third it may be helpful to perform circular dichromism with the mutants of CodY S215 to determine if they are folded properly particularly S215E which shows no binding.

4. The authors have done such thorough work in this study it is surprising they did not test the effect of the CodY S215A mutation on toxin expression.

5. Figure 9 needs statistics

**Part III – Minor Issues: Editorial and Data Presentation Modifications**

Reviewer #1: 1. A more succinct manuscript could be had if sentences on the following lines were deleted (information could be re-routed to figure legends, materials and methods, supplement, or not included at all):

a. 119-126

b. 356-361

c. 375-394 (these lines basically reiterate the results)

2. The schematics in Figures 2A, 7A, and 8A are rather simplistic and intuitive—I would suggest deleting them or moving to the supplement if you feel they are necessary. The model in Figure 10 could instead become the visual abstract instead of its own figure.

3. Fig. 2C, Could it be mentioned why recombinant PrpN migrates slower in the gel?

4. Fig. 3E, is there an understanding why a doublet band appears for PrpN in late stationary? And for Fig. 9B, is there an understanding as to why LF produces three bands in WT but only 1 predominant band in the codY S215A mutant?

Reviewer #2: Other comments:

Line 127-134. This PrpN phosphatase is present in the B. cereus group and in B. subtilis. It is not clear if this phosphatase is conserved in other Firmicutes. This point should be indicated.

Figure 2 should be a supplementary Figure.

Line 189-191 and Figure 3E: In the western blot with the anti PrpN antibody, two bands seem to be present for the point IV (late stationary phase). This point should be indicated and discussed.

Line177: the prpN mutant entered the stationary phase earlier as indicated but had also a reduced growth yield compared to the WT strain. This point should be added.

Line 216: the authors indicated that “a high population of multi-septa cells in the BAS ∆prpN strain”. It would be better to determine the proportion of cells with multiple septa. It would be more precise than a high population.

Figure 3D, 4A and 5A. The quality of the optical microscopy pictures is very poor. New figures of better quality should be provided. It is difficult to analyze the data as it is.

Role of PrpN in the sporulation process. The authors showed that prpN inactivation led to a drastic drop in spore formation. However, the sporulation does not seem to be blocked at a particular step. These results should be more carefully described and discussed.

Discussion

A recent study has shown that CodY plays a role in the control of sporulation in Bacillus anthracis (Gopalani et al, BBRC, 2016). Indeed, the overexpression of codY in B. anthracis led to a 100-fold repression of sporulation strongly suggesting that CodY is a repressor of sporulation. This reference should be cited. prpN inactivation led to a decreased sporulation efficiency and not to a sporulation derepression as expected from the increased CodY phosphorylation. This suggest that the effect of PrpN on sporulation depends on a regulator or factor downstream of CodY in the sporulation cascade. This point should be more carefully discussed.

The possible relative role of PrpC and PrpN in the B. cereus group should be discussed as well as hypothesis concerning the factors likely involved in CodY phosphorylation.

Minor points:

Line 175 : deleted “by measuring the absorbance of 600 nm until 14 hours”

Line 176: delete “to ensure axenic bacterial culture”

Line 206-207: Split the sentence. Indicate, Interestingly, these variations increased at stationary phase synchronizing with a higher PrpN production.

Line 254-260 corresponds to materials and methods. This part should be deleted. Some details could be indicated in the Figure legend if necessary.

I would suggest to replace protein expression by protein production in all the text.

Reviewer #3: 1. Figure 3 it would be helpful to show the mutants at these different time points as well.

2. Please show complement for Figure 4.

PLOS authors have the option to publish the peer review history of their article (what does this mean?). If published, this will include your full peer review and any attached files.

Reviewer #1: No

Reviewer #2: No

Reviewer #3: No
---

## [Editor Report · Decision Letter 1]

7 Jul 2022

Dear Mr. Singh,

We are pleased to inform you that your manuscript 'Role of serine/threonine protein phosphatase PrpN in the life cycle of Bacillus anthracis' has been provisionally accepted for publication in PLOS Pathogens.

Best regards,

Theresa M. Koehler

Associate Editor

PLOS Pathogens

Christoph Tang

Section Editor

PLOS Pathogens

Kasturi Haldar

Editor-in-Chief

PLOS Pathogens

orcid.org/0000-0001-5065-158X

Michael Malim

Editor-in-Chief

PLOS Pathogens

orcid.org/0000-0002-7699-2064
---

## [Editor Report · Acceptance letter]

26 Jul 2022

Dear Mr. Singh,

We are delighted to inform you that your manuscript, "Role of serine/threonine protein phosphatase PrpN in the life cycle of Bacillus anthracis," has been formally accepted for publication in PLOS Pathogens.

Best regards,

Kasturi Haldar

Editor-in-Chief

PLOS Pathogens

orcid.org/0000-0001-5065-158X

Michael Malim

Editor-in-Chief

PLOS Pathogens

orcid.org/0000-0002-7699-2064